# Guided Decoupled Exploration for Offline Reinforcement Learning Fine-tuning

## Abstract

Fine-tuning pre-trained offline Reinforcement Learning (RL) agents with online interactions is a promising strategy to improve the sample efficiency. In this work, we study the problem of sample-efficient fine-tuning for offline RL agents. Specifically, we first discussed three challenges related to the over-concentration on the offline dataset, *i.e.*, inefficient exploration, distributional shifted samples, and distorted value functions. Then, we focused on the exploration issue and investigated an important open question, *i.e.*, how to explore more efficiently in offline RL fine-tuning. Through detailed experiments, we found that it is important to relax the conservative constraints to encourage exploration while avoiding reckless actions which could ruin the learned policy. To this end, we introduced the Guided Decoupled Exploration (GDE) for fine-tuning offline RL agents, where we decouple the exploration and exploitation policies and use a dynamic teacher policy to guide exploration. Experiments on the D4RL benchmark tasks showcase the effectiveness of the proposed method.

## 1 Introduction

Offline reinforcement learning (RL) aims to learn effective policies purely from an offline dataset without online interactions with the environment (Kumar et al., 2023; Xie et al., 2023). Unlike the traditional online RL paradigm where we need to trade-off between exploration and exploitation (Sutton & Barto, 2018), offline RL completely disentangles the data collection and data inference (Riedmiller et al., 2022). In offline RL, we focus on the inference side to extract performant policies from the given dataset. Due to the demand to avoid time-consuming simulator calibration or dangerous online training, offline RL is promising for real-world RL applications (Qin et al., 2021; Xu et al., 2023).

Recently, there is a growing interest in relaxing the zero-exploration constraint in offline RL to an intermediate setting (Nair et al., 2020; Song et al., 2023; Lee et al., 2022; Wagenmaker & Pacchiano, 2022; Zhang et al., 2023a; Uchendu et al., 2023; Rafailov et al., 2023), named *Offline-to-online RL*, where we have both the access to an offline dataset and the ability to collect online samples by interacting with the environment. In this work, we follow this research direction and focus on the subproblem of fine-tuning trained offline RL agents (Zhang et al., 2023a). The goal is to use a few online samples to fine-tune an offline RL agent to improve the final performance. The main motivations can be summarized as follows: (1) The performance of the offline RL agent is limited by the quality of the offline dataset. (2) In many real-world applications, *i.e.*, robot control (Riedmiller et al., 2022) or recommender system (Bai et al., 2019; Gao et al., 2023), we can indeed deploy the RL agent to collect some online samples under specific conditions to improve the performance.

However, it is a nontrivial task to achieve sample-efficient offline RL fine-tuning due to several particular challenges. Firstly, most of the existing offline RL algorithms adopt different behavior constraints (Fujimoto & Gu, 2021; Kostrikov et al., 2021b; Wu et al., 2019; Kumar et al., 2019; Kostrikov et al., 2021a) or learn pessimistic lower-bounded $Q$-value functions (Jin et al., 2021; Li et al., 2022; Kumar et al., 2020; Yu et al., 2021) to mitigate the bootstrap error (Kumar et al., 2019), which refers to the problem of learning erroneously over-estimated $Q$-value functions by backuping *out-of-distribution* (OOD) samples (Xiao et al., 2023). Although these methods have been proven effective in extracting performant policies from offline datasets, they can also lead to some unexpected pitfalls for efficient fine-tuning, as illustrated in the following Fig 1. For example:

- *Inefficient exploration.* Model-free Offline RL policy $\pi(a|s)$ is usually biased towards insample action $a_i$ such that $\pi(a_i|s_i) > \pi(a_{OOD}|s_i)$ (Fu et al., 2022). This could lead to inefficient exploration when we use the over-conservative policy to collect online samples.

- *Distributional shifted samples.* The difference of probability density in offline dataset $\mu(s,a)$ and the online dataset $\beta(s,a)$ could lead to bootstrap error and unstable training (Lee et al., 2022).

- *Distorted value functions.* The learned value function $Q_\phi^\pi(s,a)$ in offline RL is usually far away from the optimal value function $Q^*(s,a)$ and sometimes it even diverges. Therefore, $Q_\phi^\pi(s,a)$ could be a bad parameter initialization (Nikishin et al., 2022) for later fine-tuning.

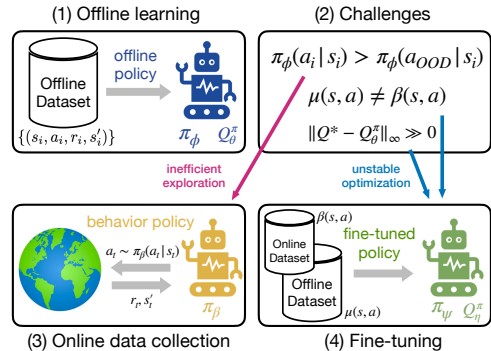

Figure 1: **Three practical challenges in the offline RL fine-tuning:** inefficient exploration, distributional shifted samples and distorted value functions.

These side effects are closely related to our concentration on the offline dataset or in essence in offline RL (Song et al., 2023; Wang et al., 2023; Hong et al., 2023). In this work, we first provide a detailed discussion of how these challenges affect fine-tuning. We then focus on addressing the exploration issue to achieve sample-efficient fine-tuning. The contributions are summarized as follows:

- We summarized three practical challenges that make efficient offline RL fine-tuning difficult.

- We introduced a general exploration method, named Guided Decoupled Exploration (GDE), that is compatible with different backbone algorithms for sample-efficient offline RL fine-tuning.

- Extensive experiments show that the proposed method outperforms other SOTA methods.

## 2 PRELIMINARIES

In this work, we consider the standard Markov Decision Process (MDP) (Puterman, 2014) setting $\mathcal{M} = (\mathcal{S}, \mathcal{A}, R, P, \rho_0, \gamma)$, where $\mathcal{S}$ and $\mathcal{A}$ are state and action spaces, $R : \mathcal{S} \times \mathcal{A} \to \mathbb{R}$ is a reward function, $P : \mathcal{S} \times \mathcal{A} \to \Delta(\mathcal{S})$ is the state-transition probability function, $\rho_0 : \mathcal{S} \to \mathbb{R}_+$ is the initial state distribution and $\gamma \in [0, 1)$ is a discount factor. Our goal is to learn a policy $\pi(a|s) : \mathcal{S} \to \Delta(\mathcal{A})$ that maximizes the expected cumulative discounted rewards $\mathbb{E}_\pi[\sum_{t=0}^\infty \gamma^t r(s_t, a_t)]$ where $s_0 \sim \rho_0$, $s_{t+1} \sim P(\cdot|s_t, a_t)$ and $a_t \sim \pi(\cdot|s_t)$.

To solve this optimization problem, value-based RL methods typically learn a state-action value function $Q^\pi(s,a) := \mathbb{E}_\pi[\sum_{t=0}^\infty \gamma^t r_t|s_0 = s, a_0 = a]$, which is defined as the expected return under policy $\pi$. For convenience, we adopt the vector notation $Q \in \mathbb{R}^{\mathcal{S} \times \mathcal{A}}$, and define the one-step Bellman operator $\mathcal{T}^\pi : \mathbb{R}^{\mathcal{S} \times \mathcal{A}} \to \mathbb{R}^{\mathcal{S} \times \mathcal{A}}$ such that $\mathcal{T}^\pi Q(s,a) := r(s,a) + \gamma \mathbb{E}_{s' \sim P, a' \sim \pi}[Q(s', a')]$. The $Q$-function $Q^\pi$ is the fixed point of $\mathcal{T}^\pi$ such that $Q^\pi = \mathcal{T}^\pi Q^\pi$ (Sutton & Barto, 2018). Similarly, we define the optimality Bellman operator as follows $\mathcal{T}Q(s,a) := r(s,a) + \gamma \mathbb{E}_{s' \sim P}[\max_{a'} Q(s', a')]$ and the optimal $Q$-value function $Q^*$ is the fixed point of $\mathcal{T}Q^* = Q^*$. In deep RL, we usually use neural networks $Q_\theta(s,a)$ to approximate the $Q$-functions by minimizing the empirical Bellman error:

$$\mathbb{E}_{(s,a,r,s')} \left[ (r + \gamma \max_{a'} Q_{\hat\theta}^\pi(s', a') - Q_\theta^\pi(s,a))^2 \right], \tag{1}$$

where we sample transitions $(s, a, r, s')$ from a replay buffer and $Q_{\hat\theta}^\pi(s,a)$ is the target network.

In offline RL (Rashidinejad et al., 2023; Touati et al., 2023), we aim to learn a policy $\pi(a|s)$ purely from a fixed offline dataset $\mathcal{D} = \{(s_i, a_i, s_i', r_i)\}$, generated by the behavior policy $\pi_\beta(a|s)$. A major challenge in offline RL is the issue of distributional shift (Xiao et al., 2023) between the learned policy $\pi(a|s)$ and the behavior policy $\pi_\beta(a|s)$. Specifically, the out-of-distribution (OOD) actions $a'$ can produce erroneously over-estimated target values for $Q_{\hat\theta}^\pi(s', a')$ in Eq (1). Therefore, many existing offline RL algorithms are motivated to constrain the learned policy to stay close to the behavior policy (Kostrikov et al., 2021b), or penalize large over-estimated $Q$-values (Fakoor et al., 2021).

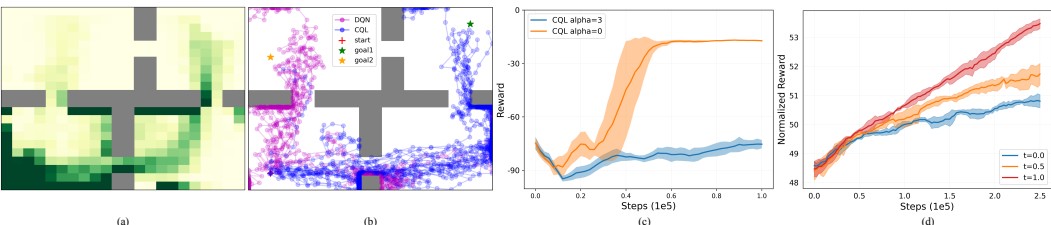

Figure 2: **The problem of conservative policy and inefficient exploration:** (a) The heatmap of the offline dataset. (b) Due to the conservative policy, the CQL agent keeps collecting similar trajectories as in the offline dataset. The DQN agent can easily find the optimal left target when we remove the CQL agent's conservative constraint. (c) Directly fine-tuning the CQL agent fails to improve the performance due to the exploration issue. (d) We can observe similar results in the continuous control task of *halfcheetah-medium-v2* when we fine-tune an IQL agent.

## 3 CHALLENGES IN FINE-TUNING OFFLINE RL AGENTS

In this section, we revisit some challenges in fine-tuning offline RL agents with experiment analysis.

### 3.1 CONSERVATIVE POLICY AND INEFFICIENT EXPLORATION

We first discuss the challenge of conservative policy and inefficient exploration. We begin with a toy maze example where an agent starts from the left bottom and attempts to reach a goal position (Fig 2). The agent receives a -1 reward at each step until it arrives at one of the two goals. We first train online DQN agents with RND bonus (Burda et al., 2018) to solve this task, and we use different checkpoints to collect trajectories to build an offline dataset. We then train an offline CQL (Kumar et al., 2020) agent on the offline dataset and then fine-tune it with further online interactions. In the experiment, we collected a dataset that is dominated by noisy trajectories and sub-optimal trajectories (ending with the right goal) on purpose. The heatmap of the collected offline data is shown in Fig 2(a). We can observe that directly fine-tuning a CQL agent can hardly improve the performance, while using a DQN agent (CQL $\alpha = 0$) can easily solve this task. From Fig 2(b), we can find that the CQL agent fails to explore the maze and keeps collecting similar trajectories as in the offline dataset.

To validate if a similar problem also exists in more complex environments with continuous action and state spaces, we further fine-tuned an IQL agent (Kostrikov et al., 2021b), one popular offline RL baseline for fine-tuning, in the *halfcheetah-medium-v2* environment. In the experiment, we sample actions from a Gaussian policy $\pi(a|s,t) = \mathcal{N}(\mu_s, t\sigma)$ with a temperature parameter $t$. From Fig 2(d), we can observe that the performance drops as we decrease the $t$ value, which indicates the negative effect caused by conservative policy and inefficient exploration.

### 3.2 DISTRIBUTIONAL-SHIFTED OFFLINE SAMPLES

We then discuss the challenge of the distribution shift issue caused by offline samples. In RL pre-training, we want to make the maximum use of prior knowledge, *i.e.*, collected samples, to improve the learning efficiency (Nair et al., 2020). However, it is still an open question of how to best use offline data during online fine-tuning (Agarwal et al., 2022). For example, simply adding offline samples to the online buffer may slow down the fine-tuning. Fig 3(a) shows the learning curves of an IQL (Kostrikov et al., 2021b), SAC (Haarnoja et al., 2018), and TD3+BC (Fujimoto & Gu, 2021) agent with and without the offline buffer during fine-tuning in the *halfcheetah-medium-v2* task. We initialize the SAC agent with parameters of an offline CQL Kumar et al. (2020) agent. We do not fine-tune the CQL agent here because it performs badly due to the poor exploration issue. Moreover, a SAC agent can be regarded as a CQL agent with conservative parameter $\alpha = 0$.

We can observe that fine-tuning an offline agent without an additional offline dataset performs significantly better in this environment. A possible reason is that the offline dataset is abundant with low-quality samples that provide limited information for the RL agent. In addition, the distribution shift between the offline dataset with probability density $\mu(s, a)$ and the online samples with

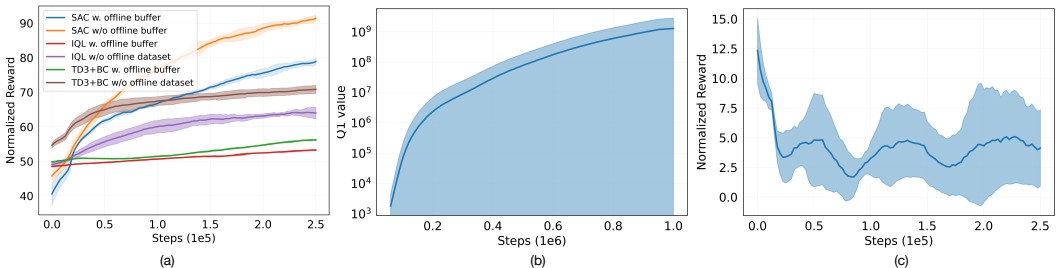

Figure 3: **Challenges of distributional-shifted offline samples and distorted value functions:** (a) Naively adding an offline dataset to the online buffer hurts the performance. (b) A bad case of fine-tuning an AWAC agent on the *walker2d-medium* environment, where the $Q$-value function diverges. (c) It is almost impossible to fine-tune such a diverged agent with further online interactions.

probability density $\beta(s, a)$, is a well-known challenge in off-policy RL that could lead to unstable training (Lee et al., 2022; Li et al., 2023a;b; Zhang et al., 2023b).

### 3.3 DISTORTED VALUE FUNCTIONS

In offline RL, the learned value functions are usually far away from the optimal value functions either due to the added conservative penalties (Kumar et al., 2020) or a failure to mitigate the bootstrap error (Kumar et al., 2019). These distorted value functions could be bad initialization parameters (Agarwal et al., 2022) that lead to slow fine-tuning or diverged results. We provide some intuitions of the side effects of distorted value functions as follows. If we set the goal of fine-tuning as learning an $\epsilon$-optimal $Q$-function such that $\|Q - Q^*\|_\infty \leq \epsilon$, then in the tabular setting the $Q$-value iteration (Agarwal et al., 2019) requires $k \geq \frac{\ln(\epsilon_0/\epsilon)}{-\ln\gamma}$ iterations to achieve $\|Q^{(k)} - Q^*\|_\infty \leq \gamma^k \|Q^{(0)} - Q^*\|_\infty \leq \epsilon$, where $\|Q^{(0)} - Q^*\|_\infty := \epsilon_0$ is the infinity norm between the offline agent's $Q$-function $Q^{(0)}$ and the optimal $Q$-value function. This equation indicates that it would be less efficient to fine-tune an offline RL agent with bad initialization. Moreover, we may encounter diverged $Q$-value functions due to bootstrap errors in offline RL. For example, Fig 3(c) shows the learning curve of fine-tuning an AWAC agent Nair et al. (2020) on the *walker2d-medium-v2* environment. As we can observe from Fig 3(b) the AWAC agent learned diverged $Q$-value functions, and it is almost impossible to fine-tune such a diverged agent using further online interactions.

### 3.4 SOME UNANSWERED KEY QUESTIONS

We end this section with some unanswered key questions. As discussed in Subsection 3.1, exploration is crucial to efficient fine-tuning. However, it is still an open question how we can explore more efficiently for fine-tuning offline RL agents (Uchendu et al., 2023)? Next, from the results in Fig 3(a), it is intriguing to ask how we can make the maximum use of the existing offline dataset (Lee et al., 2022). Another important unanswered question is whether we should select hyperparameters for fine-tuning offline RL agents. We provide more detailed discussions in the Appendix A.3.

## 4 SAMPLE EFFICIENT OFFLINE RL FINE-TUNING

In this work, we mainly concentrate on the key question of *how to explore more efficiently* in fine-tuning offline RL agents. We first conduct some preliminary experiments to shed some light on this question. Based on the experiment results, we then introduce the proposed method in Subsection 4.2.

### 4.1 HOW TO EXPLORE MORE EFFICIENTLY?

To investigate how to explore more efficiently in fine-tuning offline RL agents, we first resort to some common practices to check how effective they are. Since we now focus on the *exploration*-side, we fine-tune the offline agent only with online samples. We checked the following IQL variants.

Table 1: **Three different exploration methods:** OnIQL is the IQL baseline with only online samples.

| | OnIQL | ReIQL | RND | BpIQL |
|---|---|---|---|---|
| halfcheetah-r | **24.7 ± 2.7** | 24.7 ± 20.0 | 24.1 ± 1.8 | 19.1 ± 5.2 |
| hopper-r | 11.9 ± 0.6 | **57.4 ± 22.3** | 12.7 ± 1.1 | 11.4 ± 1.1 |
| walker2d-r | 9.0 ± 0.6 | **30.0 ± 11.2** | 10.1 ± 1.1 | 9.0 ± 0.2 |
| halfcheetah-m | 54.4 ± 1.0 | **81.6 ± 4.7** | 52.4 ± 1.0 | 51.1 ± 2.2 |
| hopper-m | **101.6 ± 0.8** | 79.7 ± 31.3 | 101.3 ± 0.5 | 58.2 ± 30.1 |
| walker2d-m | 82.1 ± 2.5 | **101.4 ± 4.6** | 83.9 ± 3.9 | 71.2 ± 16.8 |
| halfcheetah-m-re | 47.8 ± 0.5 | **70.8 ± 4.5** | 48.7 ± 0.9 | 41.7 ± 6.7 |
| hopper-m-re | 100.5 ± 3.9 | 76.5 ± 19.7 | **105.0 ± 1.2** | 86.6 ± 15.1 |
| walker2d-m-re | 94.8 ± 3.1 | **95.3 ± 15.3** | 93.0 ± 1.9 | 86.1 ± 11.7 |
| antmaze-m-p | **90.0 ± 3.3** | 70.6 ± 10.7 | 87.8 ± 1.5 | 0.0 ± 0.0 |
| antmaze-m-d | 86.2 ± 1.2 | 81.2 ± 9.7 | **91.4 ± 3.0** | 0.0 ± 0.0 |
| antmaze-l-p | 58.6 ± 8.5 | 14.2 ± 14.0 | **64.2 ± 5.6** | 0.0 ± 0.0 |
| antmaze-l-d | **67.4 ± 5.4** | 3.0 ± 6.0 | 64.6 ± 7.1 | 0.0 ± 0.0 |

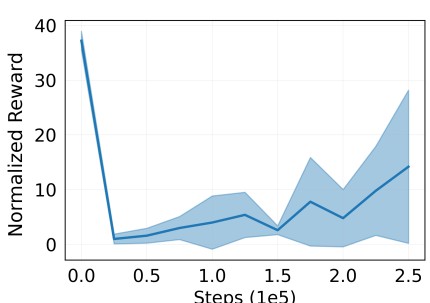

Figure 4: **Bad cases:** a crashed policy.

**Removing conservative constraints (ReIQL).** As Fig 2 shows that behavior constraints in offline RL could lead to over-conservative policies and hurt fine-tuning. Hence, we first try to simply remove such constraints during the fine-tuning. We replaced the original in-sample constrained IQL actor (Kostrikov et al., 2021b) with an unconstrained one, which maximizes the $Q$-values as in the TD3 (Fujimoto et al., 2018a) and SAC (Haarnoja et al., 2018).

**Curiosity intrinsic rewards (RND).** Using intrinsic reward (Pathak et al., 2017; Tang et al., 2017) has been shown to be an effective strategy to encourage exploration. Here, we validate the effectiveness of one of the most representative methods, *Random Network Distillation* (RND) (Burda et al., 2018), in fine-tuning offline RL agents.

**Behavioral priors (BpIQL).** Some recent works showed that learning a behavioral prior from the offline dataset with conditional generative models (Bagatella et al., 2022; Singh et al., 2021) can help to accelerate exploration. Here, we add a recently proposed method SFP (Bagatella et al., 2022), which learns a state-free prior, to the IQL agent.

From Table 1, we can observe that the ReIQL largely improves the performance in some environments in the locomotion-v2 tasks but it performs poorly in the sparse-reward antmaze-v0 tasks (Fu et al., 2020). This indicates that removing conservative constraints helps to mitigate the exploration issue in some environments where exploration is less challenging. However, such unconstrained actor could collect low-quality samples that ruin the learned policy/value functions. For example, Fig 4 shows such a policy crash issue of ReIQL in the antmaze-large-play-v0 task. We can also observe that RND performs slightly better than fine-tuning IQL with only online samples (OnIQL), which indicates that the intuition of RND is still helpful. However, the benefit of such a simple application of RND is not significant enough in our tasks. In addition, a direct application of behavioral prior performs poorly in the selected fine-tuning tasks. The main reason is that the offline dataset contains lots of low-quality samples, and we can hardly learn meaningful priors to explore the environment.

## 4.2 GUIDED DECOUPLED EXPLORATION (GDE)

### 4.2.1 MOTIVATIONS

Based on the previous discussions in Subsection 4.1, we find that none of the existing methods can achieve sample-efficient fine-tuning in all selected tasks. Moreover, we have the following key observations: (1) we can relax the conservative constraints in offline RL to encourage explorations; (2) we need to avoid reckless exploration at the early stage where bad samples could ruin the learned policy and value functions; and (3) online samples are more helpful than offline samples in the fine-tuning task. We provide more detailed discussions in the Appendix A.3.1.

Inspired by the above-mentioned observations, we develop a novel algorithm named **G**uided **D**ecoupled **E**xploration (GDE), for fine-tuning offline RL agents. In addition, we start from a *minimalist* perspective that we aim to address different challenges with simple yet effective algorithmic designs (Fujimoto & Gu, 2021). The overall pipeline of GDE is illustrated in Fig 5. The design motivation of each separate component can be summarized as follows:

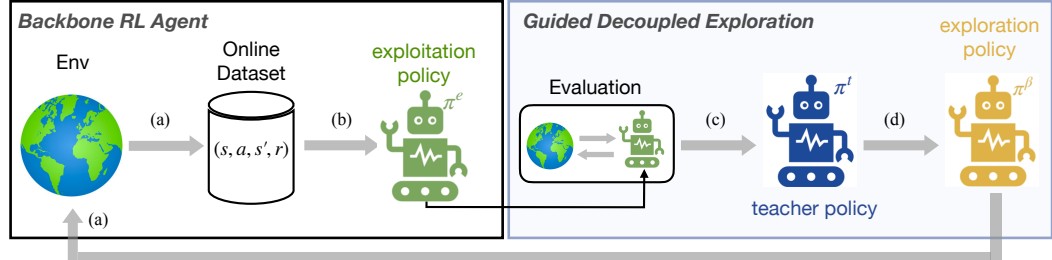

Figure 5: **The proposed method:** (a) We learn an exploration policy with a relaxed conservative constraint to collect online samples. (b) We train the exploitation policy using the backbone offline RL agent's actor loss. (c) We evaluate the exploitation policy every $F$ timesteps to update the teacher policy. (d) Teacher policy guides the exploration policy to avoid the potential policy crash issue.

- We decouple exploration and exploitation policies to relax the conservative constraint to solve the poor exploration issue, as illustrated in Fig 2.
- We adopt a dynamic teacher model to guide the exploration policy in order to avoid the potential policy crash issue, as illustrated in Fig 4.
- We follow a simple strategy that only focuses on the latest online samples and utilizes n-step return in the reward-sparse antmaze tasks to accelerate the credit assignment (Mnih et al., 2016).

### 4.2.2 BACKBONE ALGORITHM

It is notable that the proposed GDE is a general method that can be combined with different offline RL algorithms. Here, we adopt IQL (Kostrikov et al., 2021b) as the backbone algorithm due to its simplicity and SOTA performance in the fine-tuning tasks. IQL (Kostrikov et al., 2021b) is an in-sample learning algorithm which avoids any queries to values of OOD actions during the training. Specifically, IQL uses the expectile regression (ER) (Waltrup et al., 2015) to approximate the in-sample maximum $Q$-value function:

$$L_V(\psi) = \mathbb{E}\big[L_2^\tau(Q_{\hat\theta}^\pi(s,a) - V_\psi^\pi(s))\big], \tag{2}$$

where $Q_{\hat\theta}^\pi$ is the target $Q$-value function and $L_2^\tau(u) = |\tau - \mathbb{1}(u < 0)|u^2$ with hyper-parameter $\tau \in (0,1)$. Here, a larger value of $\tau < 1$ gives a better approximation to the maximum in-sample $Q$-value function. The learned $V$-function is then used to update the $Q_\theta^\pi$-value function as:

$$L_Q(\theta) = \mathbb{E}\big[(r(s,a) + \gamma V_\psi^\pi(s') - Q_\theta^\pi(s,a))^2\big], \tag{3}$$

For the policy improvement, IQL adopts the *advantage-weighted regression* loss (Peng et al., 2019):

$$L_{\pi^e}(\phi) = \mathbb{E}\big[\lambda \exp(A^\pi(s,a)) \log \pi_\phi(a|s)\big], \tag{4}$$

where $A^\pi(s,a) = Q_{\hat\theta}^\pi(s,a) - V_\psi^\pi(s)$ is the advantage and $\lambda$ is a hyper-parameter.

### 4.2.3 THE PROPOSED METHOD

A key motivation of GDE is to make minimal changes to the backbone algorithm while addressing the exploration issue in offline RL fine-tuning effectively. To achieve our objective, we explicitly maintain three policies – an exploration policy $\pi^\beta$, an exploitation policy $\pi^e$, and a teacher policy $\pi^t$. The core idea is to use a less conservative exploration policy $\pi^\beta$ to enhance data collection efficiency. The exploitation policy $\pi^e$ is responsible for policy extraction from newly collected samples. The teacher policy $\pi^t$ is dynamically updated based on the latest best exploitation policy $\pi^e$ to guide the exploration policy in sampling safe data and avoid policy crash. The exploitation policy $\pi^e$ is updated w.r.t. its default actor loss. We update the exploration policy $\pi^\beta$ by maximizing the expected $Q_\theta^\pi$ as in TD3 (Fujimoto et al., 2018a) while staying close to the teacher policy $\pi^t$ with a KL-divergence penalty. The actor loss for the exploration policy $\pi^\beta$ is as follows:

$$L_{\pi^\beta}(\mu) = -\mathbb{E}[Q_\theta^\pi(s, \pi^\beta(s))] + \rho \mathbb{D}_{KL}(\pi^t, \pi^\beta), \tag{5}$$

---

**Algorithm 1** Guided Decoupled Exploration

---

**Input:** total steps $T$, evaluation frequency $F$, trained offline RL agent $\pi_{\hat{\phi}}$, online buffer $\mathcal{D}$.

**Initialize:** exploration policy $\pi_{\mu}^{\beta}$, exploitation policy $\pi_{\phi}^{e}$, teacher policy $\pi_{\bar{\phi}}^{t}$ with $\mu, \phi, \bar{\phi} \leftarrow \hat{\phi}$.

**for** $t = 1$ **to** $T$ **do**

    Rollout $\pi_{\mu}^{\beta}$ to collect samples and add to $\mathcal{D}$.

    Update the value function by the backbone offline RL agent's critic loss.

    Update the exploitation policy $\pi_{\phi}^{e}$ by the backbone offline RL agent's actor loss.

    Update the exploration policy $\pi_{\mu}^{\beta}$ by $L_{\pi^{\beta}}(\mu) = -\mathbb{E}[Q_{\theta}^{\pi}(s, \pi^{\beta}(s))] + \rho\mathbb{D}_{KL}(\pi^{t}, \pi^{\beta})$.

    **if** $t\%F == 0$ and EvalPolicy($\pi_{\bar{\phi}}^{t}$) $<$ EvalPolicy($\pi_{\phi}^{e}$) **then**

        Update the teacher policy $\pi_{\bar{\phi}}^{t}$ by the latest exploitation policy $\bar{\phi} \leftarrow \phi$

    **end if**

**end for**

---

where the hyper-parameter $\rho$ trade-offs between active exploration and safe exploration. In the experiment, $\rho$ decays to zero with a linear schedule.

The overall pipeline of GDE is divided into two interleaving stages: a fine-tuning stage and an evaluation stage. In the fine-tuning stage, we use the exploration policy $\pi^{\beta}$ to collect samples to update the exploration policy $\pi^{\beta}$ and exploitation policy $\pi^{e}$. For every $F$ steps, we transition to the evaluation stage, employing the exploitation policy $\pi^{e}$ to to gather online samples for $L$ steps. We then assess the performance of the current exploitation policy based on cumulative rewards. The teacher policy is updated to the current exploitation policy $\pi^{e}$ if it attains the highest cumulative rewards. The collected samples in the evaluation stage are added to the buffer for later training. The total environmental steps of in the fine-tuning stage and evaluation stage contribute to the overall interaction budget. It's notable that GDE only introduced a new actor loss for exploration policy compared with the backbone offline RL agent. The pseudocode of GDE is described in Alg 1.

## 5 EXPERIMENT

In this section, we focus on the following questions: (1) How does GDE perform in the standard benchmark tasks? (2) How do different hyper-parameters affect the performance? (3) Can GDE generalize to other backbone algorithms? (4) How effective are the proposed components in GDE?

### 5.1 EXPERIMENT SETUPS

**Datasets.** We evaluate the efficacy of the proposed method on the D4RL benchmark (Fu et al., 2020). In particular, we first train the backbone algorithm on the D4RL *locomotion* and *antmaze* environments for 1M steps. Then we use different baselines to fine-tune the trained offline RL agents. Since we aim to achieve sample-efficient fine-tuning, we fine-tune the agent for 250K online steps as in (Lee et al., 2022). In all experiments, we report the average reward and standard deviation over 5 random seeds. More details of the experiment setups are described in the Appendix B.

**Baselines.** We compare to the following baselines – (1) Offline: the trained offline agent; (2) FromScratch: training a SAC (Haarnoja et al., 2018) agent from scratch; (3) AWAC (Nair et al., 2020): a prior method which is specifically designed for fine-tuning; (4) IQL (Kostrikov et al., 2021b): using the default fine-tuning method from the original paper where we add the offline dataset to the online buffer; (5) Off2OnRL (Lee et al., 2022): selecting samples w.r.t. the on-policies with density estimation; (6) JSRL (Uchendu et al., 2023): a meta-algorithm that uses a guide-policy to form different curriculums for exploration; (7) QDagger (Agarwal et al., 2022): a Reincarnating RL method which learns a policy by leveraging prior trained value functions; (8) PEX (Zhang et al., 2023a): an offline-to-online fine-tuning baseline which learns an adaptive mixture policy for tine-tuning.

### 5.2 EXPERIMENT RESULTS

Table 2 shows the experiment results on the *locomotion-v2* and *antmaze-v0* tasks, where the GDE outperforms other baselines in most tasks. We observe that most of other methods underperform the

Table 2: Evaluation results on the D4RL benchmark.

| | Offline | FromScratch | AWAC | IQL | Off2OnRL | JSRL | QDagger | PEX | GDE (Ours) |
|---|---|---|---|---|---|---|---|---|---|
| halfcheetah-r | 9.4 | $59.5 \pm 5.4$ | $30.5 \pm 2.8$ | $34.3 \pm 0.9$ | $43.3 \pm 11.0$ | $26.1 \pm 0.3$ | $27.5 \pm 1.2$ | $51.7 \pm 2.1$ | $57.6 \pm 13.4$ |
| hopper-r | 7.1 | $82.7 \pm 19.1$ | $36.6 \pm 17.6$ | $12.3 \pm 0.4$ | $40.4 \pm 24.9$ | $16.0 \pm 2.9$ | $26.7 \pm 21.7$ | $18.9 \pm 5.2$ | $92.6 \pm 7.0$ |
| walker2d-r | 5.2 | $47.7 \pm 17.8$ | $7.4 \pm 0.4$ | $9.9 \pm 0.4$ | $10.6 \pm 5.7$ | $8.2 \pm 0.3$ | $10.5 \pm 1.3$ | $9.3 \pm 0.7$ | $33.3 \pm 20.3$ |
| halfcheetah-m | 47.2 | $59.5 \pm 5.4$ | $44.4 \pm 0.1$ | $48.6 \pm 0.1$ | $47.6 \pm 2.6$ | $44.4 \pm 0.4$ | $57.5 \pm 1.2$ | $55.9 \pm 0.8$ | $\mathbf{75.7 \pm 1.8}$ |
| hopper-m | 74.7 | $82.7 \pm 19.1$ | $59.3 \pm 0.5$ | $71.7 \pm 2.2$ | $59.8 \pm 18.9$ | $41.9 \pm 2.9$ | $102.5 \pm 0.4$ | $98.1 \pm 2.6$ | $101.9 \pm 6.9$ |
| walker2d-m | 80.2 | $47.7 \pm 17.8$ | $80.5 \pm 0.3$ | $79.3 \pm 4.4$ | $66.7 \pm 24.0$ | $72.3 \pm 3.3$ | $86.9 \pm 2.3$ | $72.9 \pm 4.5$ | $\mathbf{101.7 \pm 6.1}$ |
| halfcheetah-m-re | 45.0 | $59.5 \pm 5.4$ | $41.9 \pm 0.3$ | $47.8 \pm 0.3$ | $43.8 \pm 9.9$ | $39.4 \pm 0.4$ | $48.0 \pm 0.9$ | $50.5 \pm 0.3$ | $\mathbf{70.9 \pm 1.2}$ |
| hopper-m-re | 63.8 | $82.7 \pm 19.1$ | $36.1 \pm 3.1$ | $95.6 \pm 2.5$ | $87.5 \pm 29.4$ | $46.9 \pm 2.3$ | $101.2 \pm 7.5$ | $93.8 \pm 4.2$ | $101.5 \pm 8.0$ |
| walker2d-m-re | 80.4 | $47.7 \pm 17.8$ | $73.4 \pm 1.9$ | $90.2 \pm 6.8$ | $55.0 \pm 34.7$ | $73.2 \pm 5.4$ | $100.2 \pm 4.3$ | $94.4 \pm 4.7$ | $100.9 \pm 3.9$ |
| locomotion total | 413.0 | 569.7 | 410.1 | 489.7 | 454.7 | 368.4 | 561.0 | 545.5 | **736.1** |
| antmaze-u | 85.0 | $0.0 \pm 0.0$ | $96.8 \pm 2.6$ | $94.2 \pm 1.7$ | $64.4 \pm 30.8$ | $96.8 \pm 2.5$ | $96.0 \pm 2.4$ | $97.4 \pm 1.4$ | $97.6 \pm 1.0$ |
| antmaze-u-d | 65.0 | $0.0 \pm 0.0$ | $71.2 \pm 3.2$ | $63.2 \pm 2.9$ | $49.2 \pm 24.2$ | $75.4 \pm 6.2$ | $79.6 \pm 7.7$ | $88.4 \pm 3.5$ | $90.2 \pm 2.5$ |
| antmaze-m-p | 82.0 | $0.0 \pm 0.0$ | $0.4 \pm 0.5$ | $83.6 \pm 4.1$ | $53.6 \pm 43.9$ | $89.6 \pm 1.7$ | $90.4 \pm 2.7$ | $91.0 \pm 2.8$ | $94.8 \pm 2.1$ |
| antmaze-m-d | 77.0 | $0.0 \pm 0.0$ | $1.4 \pm 0.8$ | $87.2 \pm 1.2$ | $70.8 \pm 35.0$ | $92.4 \pm 2.2$ | $89.0 \pm 2.0$ | $95.0 \pm 0.9$ | $94.4 \pm 2.1$ |
| antmaze-l-p | 39.0 | $0.0 \pm 0.0$ | $0.0 \pm 0.0$ | $49.2 \pm 3.8$ | $13.2 \pm 12.6$ | $58.2 \pm 3.2$ | $72.0 \pm 5.7$ | $54.8 \pm 6.6$ | $73.2 \pm 6.7$ |
| antmaze-l-d | 46.0 | $0.0 \pm 0.0$ | $0.2 \pm 0.4$ | $49.4 \pm 6.2$ | $10.8 \pm 13.5$ | $62.6 \pm 9.0$ | $64.8 \pm 10.1$ | $57.2 \pm 9.6$ | $80.4 \pm 2.3$ |
| antmaze total | 394.0 | 0.0 | 170.0 | 426.8 | 262.0 | 475.0 | 491.8 | 483.8 | **530.6** |
| average | 53.8 | 38.0 | 38.7 | 61.1 | 47.8 | 56.2 | 70.2 | 68.6 | **84.4** |

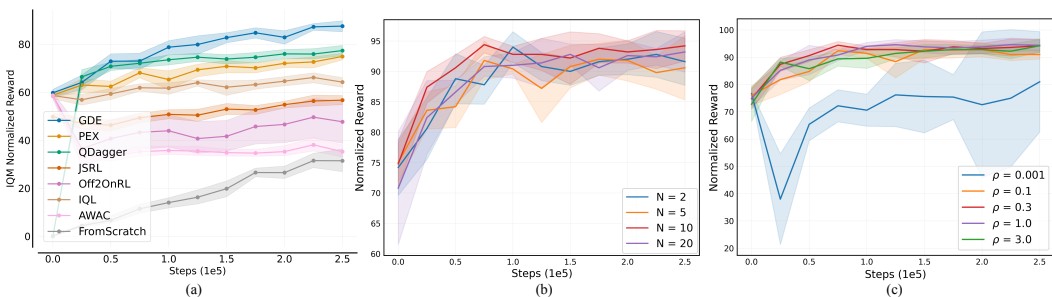

Figure 6: IQL performance and Experiments with different $N = T/F$ and $\rho$ parameters.

FromScratch baseline on the *random-v2* dataset. This is because the learned low-quality offline policy prevents the agent from effective exploration. On the contrary, GDE is able to achieve comparable results as FromScratch in two random tasks, which showcases the benefits of learning a relaxed exploration policy, especially when the initial policy is limited as in the *random-v2* task. For the more complex reward-sparse *antmaze-v0* tasks, GDE achieves the best performances in all tasks. A big challenge of the antmaze environment is that the agent could collect bad samples that ruin the policy and value functions which makes the agent fail to explore the environment effectively anymore. The superior performance of GDE in this task demonstrates that the proposed guided exploration scheme is able to achieve stable performance improvement by utilizing a dynamically updated teacher policy.

In addition, the performances of many compared methods are close to or slightly better than the Offline baseline, which shows their limitations in achieving sample-efficient fine-tuning within 250K steps. Notably, the proposed GDE outperforms the Offline baseline in two tasks by 78.2% and 34.7%, while the results of the second-best method are 35.8% and 24.8%. Moreover, Fig 6(a) shows that GDE achieves the best Interquartile Mean (IQM) performance (Agarwal et al., 2021) across all tasks. This shows that GDE is robust and performant in efficient fine-tuning across different tasks. For completeness, we also provide fine-tuning results with more online steps in Appendix C.3.

## 5.3 EFFECTS OF HYPER-PARAMETERS

We use the parameter settings for IQL (Kostrikov et al., 2021b) from the original paper. Two main hyper-parameters in GDE is the loss weight $\rho$ and the evaluation frequency $F$. Fig 6(b) and 6(c) show the effects of different $\rho$ and $F$ values on the *antmaze-medium-diverse-v0* task. For parameter $F$, we report the update number $N = T/F$, where $T$ is the max timestep. We can observe that a larger update frequency $F$ (smaller $N$) would lead to slow learning at an early stage due to the outdated teacher policy. As $\rho$ decays during the fine-tuning, the final performances for different $F$ parameters are similar. On the other hand, we can observe that GDE performs similarly w.r.t. different $\rho$ parameters except for a small value like 0.001 when the exploration policy starts to suffer from the policy crash issue.

Table 3: **Experiment of with different backbone algorithms:** the prefix T- and C- denotes TD3+BC and CQL, respectively.

| | T-Offline | T-Naive | T-QDagger | T-GDE | C-Offline | C-Naive | C-QDagger | C-GDE |
|---|---|---|---|---|---|---|---|---|
| halfcheetah-r | 11.5 | $31.5 \pm 1.3$ | $43.2 \pm 0.8$ | $\mathbf{76.5 \pm 4.3}$ | 19.0 | $17.5 \pm 7.4$ | $33.7 \pm 0.8$ | $\mathbf{38.5 \pm 0.7}$ |
| hopper-r | 8.2 | $8.1 \pm 1.4$ | $14.2 \pm 5.2$ | $18.4 \pm 6.1$ | 10.2 | $15.3 \pm 10.8$ | $13.4 \pm 0.5$ | $\mathbf{18.0 \pm 1.4}$ |
| walker2d-r | 1.1 | $9.0 \pm 5.2$ | $5.6 \pm 2.7$ | $8.2 \pm 6.7$ | 7.4 | $4.5 \pm 3.3$ | $11.0 \pm 1.0$ | $\mathbf{13.1 \pm 0.8}$ |
| halfcheetah-m | 49.6 | $50.6 \pm 0.6$ | $70.2 \pm 1.1$ | $\mathbf{77.1 \pm 2.1}$ | 45.9 | $16.6 \pm 6.9$ | $79.0 \pm 0.7$ | $\mathbf{82.9 \pm 1.3}$ |
| hopper-m | 55.4 | $30.9 \pm 4.0$ | $44.9 \pm 7.3$ | $\mathbf{104.2 \pm 1.3}$ | 67.1 | $18.4 \pm 8.5$ | $102.5 \pm 0.3$ | $102.9 \pm 0.4$ |
| walker2d-m | 79.3 | $65.5 \pm 3.5$ | $105.9 \pm 3.3$ | $103.9 \pm 4.6$ | 81.3 | $36.0 \pm 15.0$ | $103.4 \pm 4.3$ | $104.1 \pm 1.5$ |
| halfcheetah-m-re | 45.4 | $42.5 \pm 0.5$ | $65.9 \pm 1.4$ | $69.4 \pm 2.5$ | 44.6 | $43.6 \pm 3.9$ | $71.4 \pm 2.0$ | $72.6 \pm 0.5$ |
| hopper-m-re | 27.8 | $31.3 \pm 3.6$ | $35.6 \pm 3.0$ | $\mathbf{102.9 \pm 2.5}$ | 96.0 | $50.2 \pm 21.6$ | $88.9 \pm 13.6$ | $\mathbf{104.1 \pm 2.9}$ |
| walker2d-m-re | 71.2 | $65.1 \pm 7.0$ | $98.8 \pm 5.4$ | $104.1 \pm 2.8$ | 75.0 | $21.4 \pm 18.8$ | $109.2 \pm 1.8$ | $111.1 \pm 4.2$ |
| locomotion total | 349.5 | 334.5 | 484.3 | **664.7** | 446.5 | 223.5 | 612.5 | **647.3** |

## 5.4 PERFORMANCE WITH OTHER BACKBONE ALGORITHMS

In previous experiments, we select IQL as the backbone algorithm due to its simplicity and SOTA performance. Here, we evaluate GDE with other backbone algorithms. We select two model-free offline RL baselines – TD3+BC (Fujimoto & Gu, 2021) and CQL (Kumar et al., 2020). In Table 3, we compare GDE to the offline agent, naive fine-tuning and a performant baseline, QDagger. We can observe that GDE still shows strong performances with different backbone algorithms.

## 5.5 ABLATION STUDIES

Table 4 outlines the results obtained by removing different components in GDE. In the exploitation policy ablation, the exploration policy continues to be updated updated following Eqn.(5), and the teacher policy is updated based on the checkpoint of the exploration policy. In the exploration policy ablation, we uses the exploitation policy to collect samples, and we add the KL term to the exploitation policy's actor loss. For the teacher policy ablation, the KL term in the Eqn.(5) is omitted. It is evident that removing any component results in a decline in final performance. Moreover, the most impactful component varies across different tasks. For example, the exploration policy is the most important in the *locomotion-v2* tasks, while the dynamic teacher policy is the most important in the *antmaze-v0* tasks. This is because inefficient exploration is the bottleneck in *locomotion-v2* tasks, and avoiding collecting bad samples that ruin the policy is the key point in *antmaze-v0* tasks. Further, using n-step return in the antmaze tasks helps the challenging credit assignment problem. More ablation studies are discussed in the Appendix C.4.

Table 4: **Ablation studies of different components.**

| | w/o exploit | w/o explore | w/o teacher | w/o Nstep | GDE |
|---|---|---|---|---|---|
| halfcheetah-r | $41.3 \pm 28.4$ | $\mathbf{28.3 \pm 3.2}$ | $65.4 \pm 16.5$ | - | $57.6 \pm 13.4$ |
| hopper-r | $73.1 \pm 20.0$ | $\mathbf{13.0 \pm 2.0}$ | $45.6 \pm 28.2$ | - | $92.6 \pm 7.0$ |
| walker2d-r | $31.1 \pm 17.8$ | $\mathbf{9.9 \pm 1.1}$ | $38.5 \pm 8.2$ | - | $33.3 \pm 20.3$ |
| halfcheetah-m | $80.6 \pm 3.3$ | $\mathbf{56.3 \pm 1.3}$ | $84.2 \pm 4.0$ | - | $75.7 \pm 1.8$ |
| hopper-m | $\mathbf{83.6 \pm 24.0}$ | $103.2 \pm 1.2$ | $100.5 \pm 4.4$ | - | $101.9 \pm 6.9$ |
| walker2d-m | $89.5 \pm 11.8$ | $\mathbf{77.5 \pm 5.5}$ | $98.0 \pm 10.8$ | - | $101.7 \pm 6.1$ |
| halfcheetah-m-re | $77.6 \pm 2.1$ | $\mathbf{47.4 \pm 1.0}$ | $75.5 \pm 3.2$ | - | $70.9 \pm 1.2$ |
| hopper-m-re | $87.8 \pm 24.1$ | $104.2 \pm 3.2$ | $94.6 \pm 14.9$ | - | $101.5 \pm 8.0$ |
| walker2d-m-re | $92.8 \pm 14.2$ | $88.5 \pm 4.3$ | $104.4 \pm 7.4$ | - | $100.9 \pm 3.9$ |
| locomotion total | 657.4 | **528.3** | 706.7 | - | 736.1 |
| antmaze-u | $96.5 \pm 1.8$ | $95.0 \pm 3.4$ | $88.6 \pm 8.9$ | $96.0 \pm 1.5$ | $97.6 \pm 1.0$ |
| antmaze-u-d | $77.2 \pm 23.9$ | $85.5 \pm 3.5$ | $77.6 \pm 18.0$ | $55.8 \pm 18.7$ | $90.2 \pm 2.5$ |
| antmaze-m-p | $85.5 \pm 3.2$ | $87.0 \pm 4.3$ | $79.4 \pm 4.8$ | $81.2 \pm 13.5$ | $94.8 \pm 2.1$ |
| antmaze-m-d | $83.8 \pm 5.5$ | $88.8 \pm 2.8$ | $89.4 \pm 1.9$ | $91.2 \pm 1.5$ | $94.4 \pm 2.1$ |
| antmaze-l-p | $41.2 \pm 25.8$ | $70.0 \pm 5.3$ | $\mathbf{0.6 \pm 1.2}$ | $64.8 \pm 9.1$ | $73.2 \pm 6.7$ |
| antmaze-l-d | $29.2 \pm 16.9$ | $71.8 \pm 8.5$ | $\mathbf{0.0 \pm 0.0}$ | $65.2 \pm 10.5$ | $80.4 \pm 2.3$ |
| antmaze total | 413.4 | 498.1 | **335.6** | 454.2 | **530.6** |

## 6 CONCLUSION

In this paper, we investigate the problem of sample-efficient fine-tuning for offline RL agents. We begin with the introduction of three challenges in efficient offline RL fine-tuning. Through detailed experiments, we show that we need to relax the conservative constraints to encourage exploration and avoid collecting bad samples which could ruin the learned policy and value functions. Based on these key observations, we introduced the Guided Decoupled Exploration (GDE), where we decouple the exploration and exploitation policies and use a dynamic teacher policy to guide exploration. GDE shows strong performances in different benchmark tasks. An interesting future direction is to extend the fine-tuning setting to the meta RL setting. Another interesting future direction is to explore how to leverage the offline samples more effectively without slowing down the fine-tuning.

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

APPENDIX

In the appendix, we first introduce some related works and discuss the connections and differences of our work with prior works. Later, we report the details of experiment setups and additional experiment results. Lastly, we discuss the limitations and future directions of this work.

## A EXTENDED BACKGROUNDS

### A.1 RELATED WORK

**Offline RL**. Offline reinforcement learning (RL), also known as Batch RL, aims to learn performant policies purely from fixed offline datasets without any online interactions with the environment. Recent years have seen a surge of different offline RL methods, including *policy-regularized* offline RL (Fujimoto et al., 2018b; Jaques et al., 2019; Kumar et al., 2019; Wu et al., 2019; Fujimoto & Gu, 2021; Ghasemipour et al., 2021b; Kostrikov et al., 2021a; Fakoor et al., 2021), *pessimism-based* offline RL (Kumar et al., 2020; Yu et al., 2020; Buckman et al., 2020; Yu et al., 2021), *generalized behavioral cloning* methods (Nair et al., 2020; Wang et al., 2020; Chen et al., 2020; Kostrikov et al., 2021b), *model-based* offline RL (Yu et al., 2020; Kidambi et al., 2020; Yu et al., 2021; Schrittwieser et al., 2021), *uncertainty-based* offline RL (Ghasemipour et al., 2021a; Wu et al., 2021) and *ensemble-based* methods (Agarwal et al., 2020).

A large number of the prior works share a similar motivation – constraining the learned policy to stay close to the behavior policy to mitigate the issue of bootstrap error caused by the OOD actions (Fujimoto et al., 2018b; Kumar et al., 2019). Thus, various explicit or implicit constraints have been proposed, such as the KL-divergence (Jaques et al., 2019), MMD (Kumar et al., 2019), Fisher-divergence (Kostrikov et al., 2021a), or in-sample constraint (Kostrikov et al., 2021b) to regularize the learned policy. Another line of research focuses on attacking the over-estimation problem directly by using conservative penalties. For example, CQL Kumar et al. (2020) proposed a penalty regularizer for out-of-distribution (OOD) samples, such that we can learn conservative Q-functions that lower-bounds its true value. CDC Fakoor et al. (2021) introduced a new regularizer that only penalizes high-valued samples selected by the actor.

**Fine-tuning offline RL agents**. A pioneering work that discussed the problem of fine-tuning trained offline RL agents with online interactions is AWAC (Nair et al., 2020), which adopts the *advantage-weighted regression* (AWR) (Peng et al., 2019) for policy improvement. MB2PO (Villaflor et al., 2020) further introduced a two-stage method which first learns an ensemble dynamics model and then fine-tunes an AWAC agent with the model-generated samples. Later, IQL (Kostrikov et al., 2021b) which uses *exptile regression* for policy evaluation and AWR for policy improvement shows strong performance in the fine-tuning tasks. In addition, Off2OnRL (Lee et al., 2022) introduced a balanced replay scheme and pessimistic Q-ensemble to select samples by estimating the *on-policyness* and assign higher weights to the more on-policy samples. Another related work is PTR (Kumar et al., 2022) which investigated the problem of fine-tuning offline RL agents to learn new tasks in new domains for robots.

**Offline-to-online RL**. Recently, there has been a growing interest in bridging the gap between offline RL and online RL by studying an intermediate setting (Matsushima et al., 2020; Campos et al., 2021; Lee et al., 2022; Wagenmaker & Pacchiano, 2022; Uchendu et al., 2023; Agarwal et al., 2022), where we have both the access to interact with the online environment and a pre-collected offline dataset. Such an intermediate setting is much broader than the fine-tuning setting as we discussed in the main paper, where we can use a new model architecture for the online policy (Agarwal et al., 2022). For example, BREMEN (Matsushima et al., 2020) investigated the problem of low switching-cost RL which aims to achieve sample-efficient RL with limited online deployments. Behavior Transfer (BT) (Campos et al., 2021) studied the problem of using the behavior of pre-trained policies instead of using the pre-trained policy as an initialization. On the other hand, Jump-start RL (JSRL) (Uchendu et al., 2023) utilized a pre-trained guide policy to form different curriculums of starting states for the online exploration policy. Moreover, Reincarnating RL (Agarwal et al., 2022) investigated the problem of accelerating online RL by leveraging prior computations, and introduced a *policy-to-value* Reincarnating RL (PVRL) method named QDagger. Our work is also very closely related to another latest proposed method, named PEX (Zhang et al., 2023a), which maintains a policy pool and uses a mixture policy during the online finetuning.

**Theoretical Analysis**. Many recent works provide theoretical analysis for this offline-to-online RL setting. For example, HOOVI (Xie et al., 2021) provided upper and lower bounds for policy fine-tuning under different assumptions on a reference policy. Hybrid RL (Song et al., 2023) introduced a computationally and statistically efficient algorithm, called Hy-Q, when the offline dataset supports a high-quality policy and the environment has bounded bilinear rank. FineTune RL (Wagenmaker & Pacchiano, 2022) studied the problem of a number of online samples needed for fine-tuning offline policies in linear MDPs.

## A.2 CONNECTIONS AND DIFFERENCES WITH PRIOR WORKS

The proposed GDE is conceptually closely related to some prior works. Here, we discuss the connections and differences between GDE and some related works. In short, the majority of prior works discussed how to fine-tune offline RL agents, and in this work, we investigate the problem of how to fine-tune offline RL agents efficiently. More specifically, in this work, we first conduct detailed experiments in Section 4 to investigate the bottlenecks of current methods. Then we design a simple algorithm to address these bottlenecks, individually.

- PEX (Zhang et al., 2023a). A major difference between PEX and GDE is that the offline policy is fixed in PEX while we only use the offline policy as initialization in GDE. In particular, PEX uses the fixed offline policy to retain useful behaviors learned during offline training. GDE, on the other hand, adopts an adaptive teacher policy for this role. Since the offline policy is fixed in PEX, it could limit the final performance of online agents when we start from a low-quality offline policy.

- Kickstarting RL (Schmitt et al., 2018). Both GDE and Kickstarting RL adopt teacher policies. The main motivation of Kickstarting RL is to transfer knowledge from teacher policies for multi-task RL with population-based training to accelerate learning. However, GDE introduces teacher policies to guide the exploration policy to avoid the potential policy crash issue (Fig 4), where bad samples might ruin the learned policy and value functions.

- Off2OnRL (Lee et al., 2022). Both of GDE and Off2OnRL study the problem of fine-tuning trained offline RL agents with online interactions. Off2OnRL focuses on the problem of how to leverage offline samples and introduces the *Balanced Experience Replay* mechanism (which sometimes mostly selects online samples as illustrated in the Appendix A.3) On the contrary, GDE mainly focuses on the problem of how to explore more efficiently and simply uses online samples. Moreover, Off2OnRL utilized a pessimistic Q-ensemble, which not only is much more time-consuming than GDE but also underperforms GDE.

- JSRL (Uchendu et al., 2023). GDE and JSRL share a similar idea of decoupling policies to facilitate exploration. JSRL uses the offline policy to explore the environment, which we have shown to be the bottleneck in efficient fine-tuning due to over-conservatism. On the contrary, GDE explicitly removes the conservative constraint in the exploration policy to accelerate exploration.

- QDagger (Agarwal et al., 2022). Since our setting can be viewed as a special case of *Reincarnating RL*, GDE also shares some similarities with the QDagger, *i.e.*, using the teacher policy to avoid the policy crash issue. Some main differences between QDagger and GDE are that we decouple the exploration and exploitation policies to facilitate exploration and update the teacher policy dynamically to avoid the pitfalls of an outdated teacher policy.

## A.3 OPEN QUESTIONS

In this subsection, we discuss the second and the third open questions from the Subsection 3.4 in more detail.

### A.3.1 HOW TO USE THE OFFLINE DATA MORE EFFECTIVELY?

After a preliminary investigation of the efficient *exploration* problem in fine-tuning offline RL agents in Subsection 4.1, we now switch our focus to the *exploitation*-side to study the problem of how to make the maximum use of the offline dataset. Similar to the Subsection 4.1, we also use the IQL as the base agent in the case study. Instead, we now fine-tune the IQL agent with the given offline dataset.

Table 5: A comparison of different strategies to use the offline dataset: IQL naively adds offline dataset to the online buffer, Filter only uses the top 20% samples with the highest cumulative trajectory reward, PER uses the standard TD error and 5-step returns, and Off2OnRL directly estimates the density ratio of each sample.

|  | IQL | Filter | PER | Off2OnRL |
|---|---|---|---|---|
| halfcheetah-r | $34.3 \pm 0.9$ | $\mathbf{47.4 \pm 0.9}$ | $31.8 \pm 1.1$ | $43.3 \pm 11.0$ |
| hopper-r | $12.3 \pm 0.4$ | $10.9 \pm 1.2$ | $14.2 \pm 4.2$ | $\mathbf{40.4 \pm 24.9}$ |
| walker2d-r | $9.9 \pm 0.4$ | $10.5 \pm 0.5$ | $9.6 \pm 0.4$ | $\mathbf{10.6 \pm 5.7}$ |
| halfcheetah-m | $48.6 \pm 0.1$ | $\mathbf{51.0 \pm 0.2}$ | $48.1 \pm 0.1$ | $47.6 \pm 2.6$ |
| hopper-m | $71.7 \pm 2.2$ | $\mathbf{83.7 \pm 5.7}$ | $65.6 \pm 5.8$ | $59.8 \pm 18.9$ |
| walker2d-m | $79.3 \pm 4.4$ | $73.2 \pm 2.7$ | $\mathbf{82.1 \pm 3.7}$ | $66.7 \pm 24.0$ |
| halfcheetah-m-re | $47.8 \pm 0.3$ | $\mathbf{50.3 \pm 0.2}$ | $45.2 \pm 0.3$ | $43.8 \pm 9.9$ |
| hopper-m-re | $\mathbf{95.6 \pm 2.5}$ | $92.9 \pm 7.2$ | $88.7 \pm 5.7$ | $87.5 \pm 29.4$ |
| walker2d-m-re | $90.2 \pm 6.8$ | $96.8 \pm 3.8$ | $\mathbf{99.6 \pm 1.7}$ | $55.0 \pm 34.7$ |
| locomotion total | 489.7 | **516.7** | 484.9 | 454.7 |
| antmaze-u | $\mathbf{94.2 \pm 1.7}$ | $93.4 \pm 2.1$ | $87.4 \pm 4.5$ | $64.4 \pm 30.8$ |
| antmaze-u-d | $\mathbf{63.2 \pm 2.9}$ | $48.4 \pm 13.4$ | $36.8 \pm 21.5$ | $49.2 \pm 24.2$ |
| antmaze-m-p | $83.6 \pm 4.1$ | $\mathbf{83.8 \pm 9.2}$ | $79.4 \pm 2.2$ | $53.6 \pm 43.9$ |
| antmaze-m-d | $87.2 \pm 1.2$ | $\mathbf{89.8 \pm 1.6}$ | $86.6 \pm 3.2$ | $70.8 \pm 35.0$ |
| antmaze-l-p | $\mathbf{49.2 \pm 3.8}$ | $28.0 \pm 6.8$ | $18.6 \pm 8.6$ | $13.2 \pm 12.6$ |
| antmaze-l-d | $49.4 \pm 6.2$ | $\mathbf{53.2 \pm 6.5}$ | $37.0 \pm 7.5$ | $10.8 \pm 13.5$ |
| antmaze total | **426.8** | 396.6 | 345.8 | 262.0 |

As shown in Fig 3(a), simply adding offline samples to the online buffer during the fine-tuning is harmful in the *halfcheetah-medium-v2* task. In Table 5, we list the results of fine-tuning an IQL agent with an offline dataset on more tasks. Compared with the results of OnIQL in Table 1, we can observe that IQL usually underperforms OnIQL except for the *random* tasks, which suggests that we need to search for a better strategy to leverage the offline dataset.

**Filtering low-quality samples.** Inspired by some prior works in offline imitation learning (Chen et al., 2020), we first try to filter out low-quality samples in the offline dataset. Here, we use the cumulative trajectory reward as the proxy to evaluate the quality of an offline sample. For each offline dataset, we only keep the top 20% samples with the highest cumulative trajectory rewards. As Table 5 shows the Filter-version of IQL outperforms standard IQL in some tasks, but the overall performance gain is not significant.

**Prioritized experience replay.** We next compare to the popular non-uniform sampling scheme, *prioritized experience replay* (PER), where we follow the use of temporal-difference (TD) error as the priority metric (Schaul et al., 2015). In addition, we use an n-step return with horizon 5 as it has shown to be effective in multiple off-policy RL methods (Fedus et al., 2020). However, as Table 5 shows PER with TD error is less effective in our problem. The main reason is that it usually selects offline samples with large TD errors and slows the learning from online samples.

**Probability density estimation.** We finally compare to a recently proposed method, named Off2OnRL (Lee et al., 2022), which directly estimates the probability density of each sample in the replay buffer (Sinha et al., 2022) and selects samples according to the estimated *on-policyness*. However, results in Table 5 show that Off2OnRL suffers from high variance due to the unstable density estimation.

### A.3.2 HOW TO SELECT HYPER-PARAMETERS?

During the experiments, we found that fine-tuning offline RL agents could suffer from large variance, especially when the offline agent is trained with low-quality datasets. For example, we plot the learning curves across different random seeds on the *hopper-random-v2* and *walker2d-random-v2* tasks in the Fig 7. Such a large variance could be due to parameter initializations or hyper-parameter selections. For example, when Off2OnRL has a *good* parameter initialization, it would mostly select online samples (upper right) and achieve good performances in the *hopper-random-v2* task. However, hyper-parameter selection (Paine et al., 2020; Zhang & Jiang, 2021) and off-policy policy evaluation (OPE) (Konyushova et al., 2021) for offline RL agents could itself be a challenging open question. We leave this problem for future works.

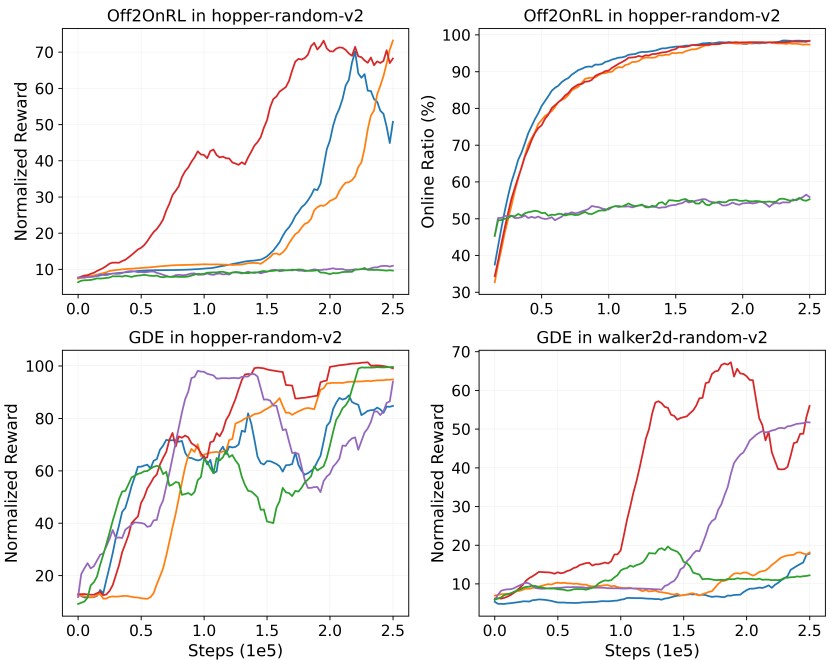

Figure 7: Learning curves on the *hopper-random-v2* and *walker2d-random-v2* across different random seeds. We can observe that fine-tuning offline RL agents sometimes suffers from large variance, which could be due to parameter initializations or hyper-parameter selections. For example, when Off2OnRL has a *good* parameter initialization, it would mostly select online samples (upper right) and achieve good performances in the *hopper-random-v2* task.

## B  EXPERIMENT DETAILS

In this section, we present more details about the dataset and the baseline algorithms. Then we summarize the details of each experiment in the main paper.

### B.1  EXPERIMENT SETUPS

In the experiment, we evaluate all baseline agents on the standard D4RL benchmark (Fu et al., 2020). In specific, we use the "-v2" version dataset, which contains more metadata and fixes bugs in the "-v0" version dataset [1], to train the offline RL agent for 1M steps. In all experiments, we report the average result and standard deviation over 5 random seeds. We run all experiments on a workstation with a GeForce GTX 3090 GPU and an Intel Core i9-12900KF CPU. We report the average result of the last 10 evaluations for the *locomotion-v2* tasks and the last evaluation result for the *antmaze-v0* tasks. We summarize the tasks used for the experiments in Table 6.

### B.2  BASELINE ALGORITHMS

In the experiment, we re-implement the IQL (Kostrikov et al., 2021b), TD3+BC (Fujimoto & Gu, 2021) and CQL (Kumar et al., 2020) and different baselines in JAX (Frostig et al., 2018). For the software, we use the following versions:

- Python 3.9
- Jax 0.4.1
- Gym 0.21.0

---

[1] https://github.com/rail-berkeley/d4rl/wiki/Tasks

Table 6: Tasks and the corresponding environment names in the D4RL benchmark.

| Task | D4RL Environment Name |
|---|---|
| antmaze-u | antmaze-umaze-v0 |
| antmaze-u-d | antmaze-umaze-diverse-v0 |
| antmaze-m-p | antmaze-medium-play-v0 |
| antmaze-m-d | antmaze-medium-diverse-v0 |
| antmaze-l-p | antmaze-large-play-v0 |
| antmaze-l-d | antmaze-large-diverse-v0 |
| halfcheetah-r | halfcheetah-random-v2 |
| hopper-r | hopper-random-v2 |
| walker2d-r | walker2d-random-v2 |
| halfcheetah-m | halfcheetah-medium-v2 |
| hopper-m | hopper-medium-v2 |
| walker2d-m | walker2d-medium-v2 |
| halfcheetah-m-re | halfcheetah-medium-replay-v2 |
| hopper-m-re | hopper-medium-replay-v2 |
| walker2d-m-re | walker2d-medium-replay-v2 |

Table 7: Main hyper-parameters for GDE in the experiment.

| Hyper-parameters | Values |
|---|---|
| expectile value $\tau$ | 0.9 |
| advantage temperature $\lambda$ | 10.0 |
| learning rate | 3e-4 |
| replay buffer size | 2.5e5 |
| batch size | 256 |
| discount factor $\gamma$ | 0.99 |
| nstep horizon | 3 |
| KL loss weight $\rho$ | 0.3 |
| update frequency $F$ | 2.5e4 |
| evaluation trajectory length | 5000 |

- Mujoco 2.1.2
- mujoco-py 2.1.2.14
- numpy 1.21.1
- d4rl 1.1

For Off2OnRL, we use the density estimation and balanced replay from the official implementation [2]. For SFP, we adopt the state-free prior from the official implementation [3].

### B.3 HYPER-PARAMETERS

We summarize the main hyper-parameters for GDE for the *antmaze-v0* in the Table 7. Most of the parameters are taken from the original IQL paper (Kostrikov et al., 2021b). For the *locomotion-v2* tasks, we use $\tau = 0.7$, $\lambda = 3.0$ from the IQL paper, and set $\rho$=0.3 and $F$=2.5e3. In the experiment, all models are implemented using 2-layer MLPs with hidden dimension 256. For the SAC baseline, we set the target entropy to be the negative action dimensions. For the CQL baseline, we set the conservative $\alpha$ to be 5.0. For the JSRL baseline, we set the curriculum stages to be 10. For the QDagger baseline, we set the distillation loss weight to be 0.3. For the PEX baseline, we use the same hyperparameters as IQL.

---

[2]https://github.com/shlee94/Off2OnRL
[3]https://github.com/eth-ait/sfp

### B.4 TOY EXAMPLE

In the toy example (Fig 2), we first train an online DQN agent with/without RND (Burda et al., 2018) exploration bonus for 80K timesteps. Then we intentionally use different checkpoints to collect an offline dataset that consists of more noisy/sub-optimal trajectories than optimal trajectories. The heatmap of the dataset is plotted in the Fig 2(a). The offline CQL agent can learn a sub-optimal policy to reach the right goal with around 70 steps. Fig 2(b) shows the performance during the online fine-tuning, where CQL $\alpha$ is the conservative parameter and $\alpha = 0$ corresponds to the vanilla DQN agent. We set $\alpha = 3$ during the training of the offline CQL agent. We can observe that using the CQL agent to interact with the environment directly fails to improve the performance, while a DQN agent with the same parameter initialization can easily solve this task. Such a performance gap is due to the poor exploration ability of the CQL agent. Fig 2(c) plots the trajectories of agents during the fine-tuning, and we can observe that the CQL agent keeps collecting similar transitions as in the offline dataset.

### B.5 A BRIEF SUMMARY OF GDE AND THE THREE CHALLENGES

Here, we provide a brief summary of the proposed GDE method and the three challenges in efficient offline RL fine-tuning. Firstly, the three challenges, *i.e.*, inefficient exploration, distributional shifted samples, and distorted value functions, are all closely related to our over-concentration of the offline dataset during training of the offline RL agent. The proposed GDE method addressed the inefficient exploration issue (Fig 2) by decoupling the exploration and exploitation policies. In particular, GDE used the original policy from the backbone algorithm as the exploitation policy for policy improvement and used the less conservative exploration policy to collect online samples. In order to avoid the potential policy crash issue (Fig 4), we further introduced a teacher policy, which is updated by the latest best-performing exploitation policy, to guide the exploration policy to sample safe actions. We use a simple yet effective trick to fine-tune with online samples to mitigate the challenge of distributional shifted samples. The distorted value functions exist in offline RL fine-tuning as long as we use the value functions from the trained offline agent. We can mitigate the distorted value function issue by selecting a good checkpoint model without diverged value functions.

We then provide some theoretical insights about the three challenges and inefficient fine-tuning. Firstly, the inefficient exploration issue can be explained by the Corollary 4.5 from (Kakade & Langford, 2002):

$$\eta_D(\pi^*) - \eta_D(\pi) \leq \frac{\epsilon}{1 - \gamma} \left\| \frac{d_{\pi^*, D}}{d_{\pi, \mu}} \right\|_\infty,$$

where $\eta_\mu(\pi) = \mathbb{E}_{s \sim \mu}[V^\pi(s)]$ is the weighted value function of policy $\pi$ with a distribution $\mu$, $d_{\pi^*, D}$ is the state distribution of the optimal policy, and $d_{\pi, \mu}$ is the state distribution of policy $\pi$. This shows that the upper bound of the difference between optimal policy and policy $\pi$ is proportional to the mismatch between the state distribution of states of current policy and optimal policy $\left\| \frac{d_{\pi^*, D}}{d_{\pi, \mu}} \right\|_\infty$. A state distribution $d_{\pi, \mu}$ that is closer to the optimal state distribution helps the performance. In the offline RL fine-tuning, when the policy is over-conservative and fails to explore the environment efficiently, then $\left\| \frac{d_{\pi^*, D}}{d_{\pi, \mu}} \right\|_\infty$ is large, hence leading to inefficient fine-tuning. Secondly, the *distribution shift* issue in off-policy RL is a well-known challenge as discussed in the *deadly triad* problem (Van Hasselt et al., 2018; Fu et al., 2019). Combining the function approximation, TD-learning, and off-policy data does always converge. Lastly, if we set our goal as to learn a near-optimal $Q$-function in the fine-tuning task. We denote the initialized offline $Q$-function as $Q^{(0)}$ and $Q^*$ as the optimal value function, then in the tabular setting the $Q$-value iteration (Agarwal et al., 2019), at the infinity norm between $Q^{(k)}$ and $Q^*$ at $k$-th iteration is:

$$\|Q^{(k)} - Q^*\|_\infty = \|\mathcal{T}^k Q^{(0)} - \mathcal{T}^k Q^*\|_\infty \leq \gamma^k \|Q^{(0)} - Q^*\|_\infty.$$

If we want to learn an $\epsilon$-optimal value function such that $\|Q^{(k)} - Q^*\|_\infty \leq \epsilon$, then we need $k \geq \frac{\ln(\epsilon_0/\epsilon)}{-\ln \gamma}$ iterations. This illustrates the the issue of distorted value function in offline RL fine-tuning, where a bad initialization of $Q^{(0)}$ can slow the convergence. In practice, we found that a diverged offline $Q$-function can hardly be fine-tuned, hence, we only select good offline checkpoint models without diverged $Q$-functions.

## B.6 Loss Functions for BpIQL

In Table 1, we compare to a behavior prior-based IQL variant (BpIQL) to study the effect of using a learned behavior prior to accelerating exploration. In the experiment, we adopt the latest proposed SFP algorithm, which learns a state-free prior (Bagatella et al., 2022). Notably, the original SFP algorithm uses SAC (Haarnoja et al., 2018) as the base agent, while we use the IQL agent in the experiment. Here, we provide the modified objective functions for BpIQL for completeness. In the origin IQL algorithm, objective functions for the value network and critic network are as follows:

$$L_V(\psi) = \mathbb{E}_{(s,a)\sim\mathcal{D}}[L_2^\tau(Q_{\hat{\theta}}(s,a) - V_\psi(s))],$$
$$L_Q(\theta) = \mathbb{E}_{(s,a,s')\sim\mathcal{D}}[(r(s,a) + \gamma V_\psi(s') - Q_\theta(s,a))^2].$$

In SFP (Bagatella et al., 2022), we use a mixture policy that samples from the behavior prior $\bar{\pi}(a_t|H_t)$ with probability $\lambda_t$ and samples from the current policy $\pi(a_t|s_t)$ with probability $1 - \lambda_t$, where $H_t$ is the trajectory history.

$$\tilde{\pi} = \lambda_t\bar{\pi}(a_t|H_t) + (1 - \lambda_t)\pi(a_t|s_t).$$

We can express the value function recursively as $Q_\tau(s,a) = r(s,a) + \gamma\mathbb{E}_{s'\sim p(\cdot|s,a)}[V_\tau(s')]$. Then the value function of the mixture function can be expressed as:

$$V_\tau^{\tilde{\pi}}(s) = \mathbb{E}_{\tilde{a}\sim\tilde{\pi}(\cdot|s)}^\tau[Q_\tau^{\tilde{\pi}}(s,\tilde{a})] = \int_{\tilde{a}} Q_\tau^{\tilde{\pi}}(s,\tilde{a})\tilde{\pi}(\tilde{a}|s)$$
$$= \lambda\int_{\bar{a}} Q_\tau^{\tilde{\pi}}(s,\bar{a})\bar{\pi}(\bar{a}) + (1-\lambda)\int_a Q_\tau^{\tilde{\pi}}(s,a)\pi(a|s)$$
$$= \lambda\mathbb{E}_{\bar{a}\sim\bar{\pi}}[Q_\tau^{\tilde{\pi}}(s,\bar{a})] + (1-\lambda)\mathbb{E}_{a\sim\pi}[Q_\tau^{\tilde{\pi}}(s,a)],$$

where $\lambda = \Lambda(s)$ is the mixing weight. The $Q$-function for the mixture policy $\tilde{\pi}$ is:

$$Q_\tau^{\tilde{\pi}}(s,a) = r(s,a) + \gamma\mathbb{E}_{s'\sim p(\cdot|s,a)}[V_\tau^{\tilde{\pi}}(s')]$$
$$= r(s,a) + \gamma\mathbb{E}_{s'\sim p(\cdot|s,a)}[\lambda\mathbb{E}_{\bar{a}'\sim\bar{\pi}}Q_\tau^{\tilde{\pi}}(s',\bar{a}') + (1-\lambda)\mathbb{E}_{a'\sim\pi}Q_\tau^{\tilde{\pi}}(s',a')]$$

We later optimize the parameter $\omega$ by maximizing $V^{\tilde{\pi}}(s)$:

$$\max_\omega V^{\tilde{\pi}}(s) = \max_\omega \Lambda_\omega(s)\mathbb{E}_{\bar{a}\sim\bar{\pi}}[Q_\tau^{\tilde{\pi}}(s,\bar{a})] + (1 - \Lambda_\omega(s))\mathbb{E}_{a\sim\pi}[Q_\tau^{\tilde{\pi}}(s,a)]$$
$$= \max_\omega \Lambda_\omega(s)\mathbb{E}_{\bar{a}\sim\bar{\pi},a\sim\pi}[Q_\tau^{\tilde{\pi}}(s,\bar{a}) - Q_\tau^{\tilde{\pi}}(s,a)].$$

## B.7 Experiment Learning Curves

Fig 8 plots the learning curves on the *antmaze-v0* tasks.

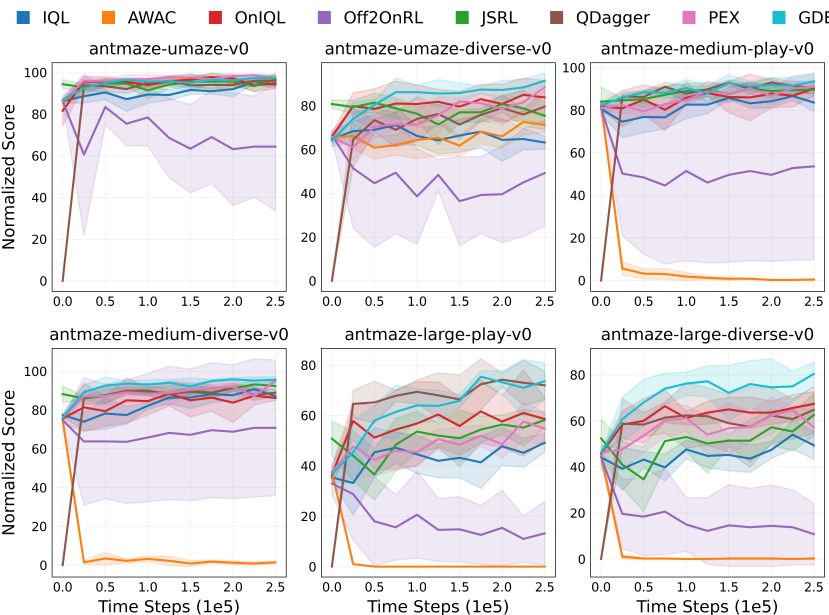

Figure 8: Learning curves on the *antmaze-v0* tasks.

Fig 9 plots the learning curves on the *locomotion-v2* tasks. We use a sliding window of length 11 to smooth the curves to facilitate visualization.

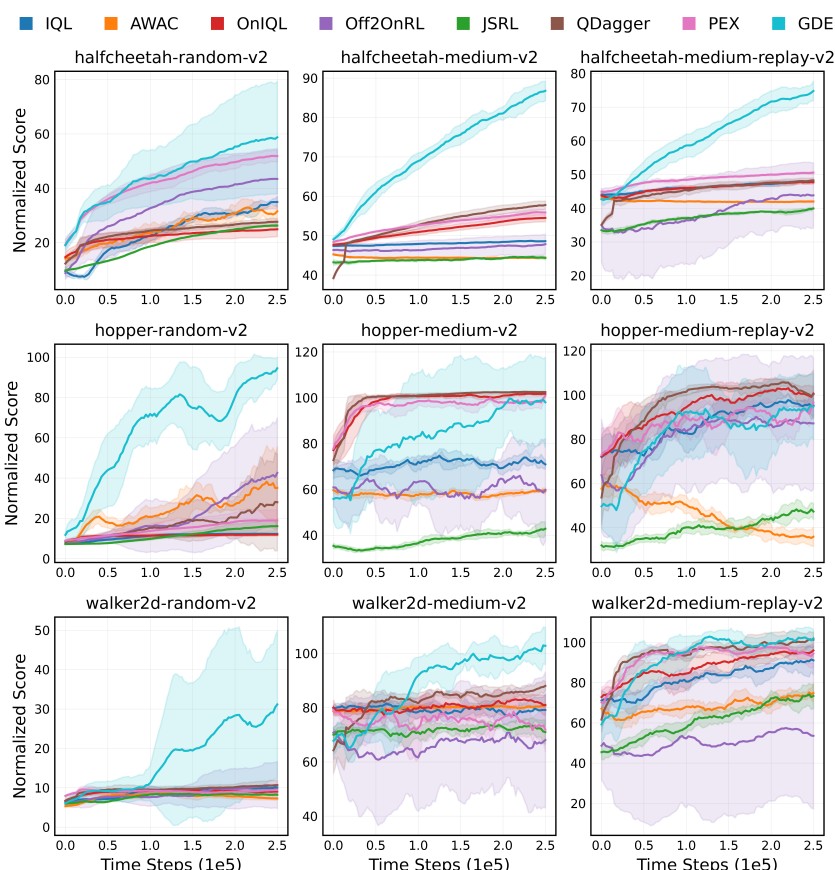

Figure 9: Learning curves on the *locomotion-v2* tasks.

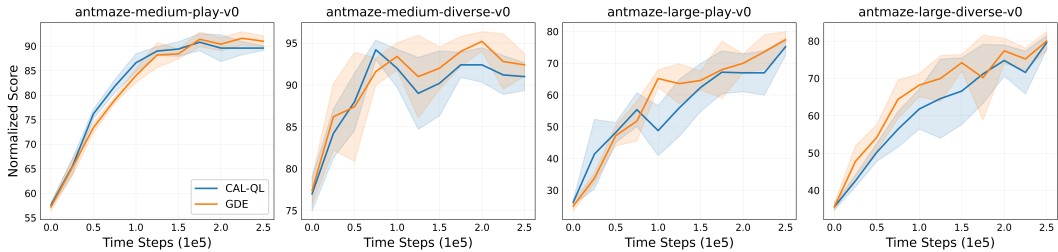

Figure 10: Comparison of GDE and CAL-QL.

# C    ADDITIONAL EXPERIMENTS

## C.1    COMPARISON WITH SAC WITH PRELOAD BUFFER

In this subsection, we compare GDE to training an off-policy algorithm, SAC (Haarnoja et al., 2018), with a preload buffer. From Table 8, we can observe that using a preload buffer sometimes hurts the performance when the data quality is low, *i.e.*, in the random tasks. Moreover, training a SAC from scratch failed to solve the antmaze tasks no matter we use a preload buffer or not.

Table 8: Compare to SAC with a preload buffer.

|  | SAC without buffer | SAC with buffer | GDE |
|---|---|---|---|
| halfcheetah-r | 59.5 (5.4) | 52.4 (2.4) | $57.6 \pm 13.4$ |
| hopper-r | 82.7 (19.1) | 25.6 (4.0) | $92.6 \pm 7.0$ |
| walker2d-r | 47.4 (17.8) | 15.4 (8.1) | $33.3 \pm 20.3$ |
| halfcheetah-m | $59.5 \pm 5.4$ | 61.1 (0.8) | $\mathbf{75.7 \pm 1.8}$ |
| hopper-m | $82.7 \pm 19.1$ | 66.6 (11.2) | $\mathbf{101.9 \pm 6.9}$ |
| walker2d-m | $47.7 \pm 17.8$ | 73.0 (11.2) | $\mathbf{101.7 \pm 6.1}$ |
| halfcheetah-m-re | $59.5 \pm 5.4$ | 67.5 (1.8) | $70.9 \pm 1.2$ |
| hopper-m-re | $82.7 \pm 19.1$ | 85.1 (19.4) | $101.5 \pm 8.0$ |
| walker2d-m-re | $47.7 \pm 17.8$ | 84.0 (1.1) | $\mathbf{100.9 \pm 3.9}$ |
| locomotion total | 569.7 | 530.7 | **736.1** |
| antmaze-u | $0.0 \pm 0.0$ | 2.4 (0.5) | $\mathbf{97.6 \pm 1.0}$ |
| antmaze-u-d | $0.0 \pm 0.0$ | $0.0 \pm 0.0$ | $\mathbf{90.2 \pm 2.5}$ |
| antmaze-m-p | $0.0 \pm 0.0$ | $0.0 \pm 0.0$ | $\mathbf{94.8 \pm 2.1}$ |
| antmaze-m-d | $0.0 \pm 0.0$ | $0.0 \pm 0.0$ | $\mathbf{94.4 \pm 2.1}$ |
| antmaze-l-p | $0.0 \pm 0.0$ | $0.0 \pm 0.0$ | $\mathbf{73.2 \pm 6.7}$ |
| antmaze-l-d | $0.0 \pm 0.0$ | $0.0 \pm 0.0$ | $\mathbf{80.4 \pm 2.3}$ |
| antmaze total | 0.0 | 2.4 | **530.6** |

## C.2    COMPARISON WITH CAL-QL

In this subsection, we compare GDE to the CAL-QL (Nakamoto et al., 2023). It is notable that CAL-QL is a specific two-stage offline RL fine-tuning algorithm, while GDE is a general offline RL fine-tuning framework. CAL-QL first uses a modified CQL loss to learn an offline RL agent, and then fine-tune this agent with a mixture of offline and online samples. However, GDE is agnostic to the offline learning stage and can be combined with any other backbone offline RL agents. Here, we compare CAL-QL and GDE on the antmaze tasks in Fig 10. We can observe that GDE is also effective when we use a CAL-QL agent as the backbone algorithm.

## C.3    FINE-TUNING WITH 1M STEPS

In the main paper, we mainly focus on sample-efficient fine-tuning for offline RL agents. Therefore, we report the results with 250K steps online interactions in Table 2. Here, we report the results with 1M steps of online interactions for completeness. We compare the proposed GDE with the two best baselines, OnIQL and QDagger, in the previous experiments. Table 9, Fig 11 and Fig 12 show the results of fine-tuning agents with 1M steps. We can observe that GDE still show strong performances w.r.t. longer fine-tuning steps.

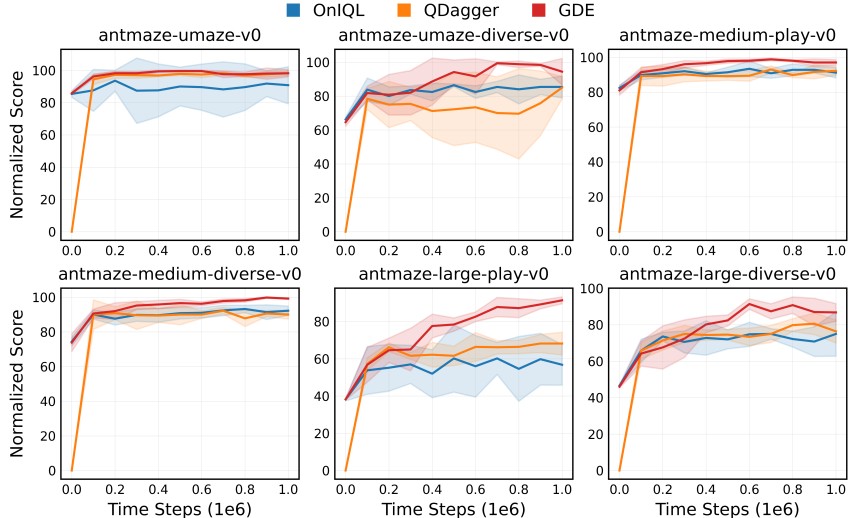

Figure 11: Fine-tuning with 1M steps on the *antmaze-v0* tasks.

Table 9: Fine-tuning with 1M steps.

|  | OnIQL (250K) | OnIQL (1M) | QDagger (250K) | QDagger (1M) | GDE (250K) | GDE (1M) |
|---|---|---|---|---|---|---|
| halfcheetah-r | 24.7 ± 2.7 | 33.2 ± 4.1 | 27.5 ± 1.2 | 40.6 ± 1.1 | 57.6 ± 13.4 | 95.3 ± 6.8 |
| hopper-r | 11.9 ± 0.6 | 18.2 ± 7.6 | 26.7 ± 21.7 | 38.6 ± 21.3 | 92.6 ± 7.0 | 96.1 ± 13.7 |
| walker2d-r | 9.0 ± 0.6 | 11.9 ± 2.4 | 10.5 ± 1.3 | 14.5 ± 4.4 | 33.3 ± 20.3 | 86.0 ± 11.0 |
| halfcheetah-m | 54.4 ± 1.0 | 58.2 ± 2.1 | 57.5 ± 1.2 | 60.0 ± 0.4 | 75.7 ± 2.4 | 96.8 ± 0.8 |
| hopper-m | 101.6 ± 0.8 | **106.7 ± 1.4** | 102.5 ± 0.4 | 102.2 ± 0.2 | 101.9 ± 6.9 | 104.2 ± 8.8 |
| walker2d-m | 82.1 ± 2.5 | 83.0 ± 4.6 | 86.9 ± 2.3 | 90.4 ± 3.0 | 101.7 ± 6.1 | 111.4 ± 1.2 |
| halfcheetah-m-re | 47.8 ± 0.5 | 48.9 ± 1.0 | 48.0 ± 0.9 | 53.4 ± 1.3 | 70.9 ± 1.2 | 92.2 ± 1.6 |
| hopper-m-re | 100.5 ± 3.9 | 102.4 ± 8.1 | 101.2 ± 7.5 | **106.6 ± 2.7** | 101.5 ± 8.0 | 98.6 ± 11.5 |
| walker2d-m-re | 94.8 ± 3.1 | 99.6 ± 3.6 | 100.2 ± 4.3 | 102.3 ± 1.6 | 100.9 ± 3.9 | 114.1 ± 4.2 |
| locomotion total | 526.8 | 562.1 | 561.0 | 608.6 | 736.1 | 904.7 |
| antmaze-u | 94.8 ± 2.4 | 90.8 ± 11.5 | 96.0 ± 2.4 | **98.2 ± 1.5** | 97.6 ± 1.0 | 98.2 ± 2.2 |
| antmaze-u-d | 83.8 ± 7.7 | 85.4 ± 6.4 | 79.6 ± 7.7 | 85.0 ± 6.6 | 90.2 ± 2.5 | 98.2 ± 1.5 |
| antmaze-m-p | 90.0 ± 3.3 | 91.2 ± 3.0 | 90.4 ± 2.7 | 92.2 ± 2.5 | 94.8 ± 2.1 | 97.0 ± 1.9 |
| antmaze-m-d | 86.2 ± 1.2 | 92.2 ± 2.7 | 89.0 ± 2.0 | 90.0 ± 2.6 | 94.4 ± 2.1 | 99.2 ± 0.4 |
| antmaze-l-p | 58.6 ± 8.5 | 56.8 ± 10.9 | 72.0 ± 5.7 | 68.2 ± 6.1 | 73.2 ± 6.7 | 91.5 ± 2.1 |
| antmaze-l-d | 67.4 ± 5.4 | 75.0 ± 12.2 | 64.8 ± 10.1 | 76.4 ± 6.3 | 80.4 ± 2.3 | 88.0 ± 5.0 |
| antmaze total | 480.8 | 491.4 | 491.8 | 510.0 | 530.6 | 572.1 |

## C.4 MORE ABLATION STUDIES

We also conduct more ablation studies w.r.t. the n-step return in the *antmaze-v0* tasks. Table 10 and Fig 13 show the results with different n-step horizons. We can observe that a horizon of 3 achieves the best performance, where a large horizon would incur higher variances in the challenging *antmaze-large* environments and the single-step return is less efficient to solve the difficult credit assignment problem.

Table 10: Experiments with different n-step return.

|  | n=1 | n=3 | n=5 | n=10 |
|---|---|---|---|---|
| antmaze-u | 94.6 ± 3.6 | 97.4 ± 1.0 | 98.2 ± 1.3 | **98.6 ± 2.0** |
| antmaze-u-d | 58.4 ± 18.1 | 91.4 ± 3.6 | **93.6 ± 3.4** | 89.4 ± 6.6 |
| antmaze-m-p | 83.0 ± 12.6 | 93.6 ± 3.4 | 93.8 ± 3.1 | **95.0 ± 2.4** |
| antmaze-m-d | 91.0 ± 1.4 | 95.6 ± 1.6 | **97.0 ± 1.7** | 94.0 ± 2.6 |
| antmaze-l-p | 65.8 ± 8.4 | **73.6 ± 7.3** | 73.0 ± 9.1 | 62.6 ± 15.2 |
| antmaze-l-d | 68.6 ± 11.5 | **80.4 ± 5.2** | 74.2 ± 4.3 | 77.4 ± 9.0 |
| antmaze total | 461.3 | **532.0** | 529.8 | 517.0 |

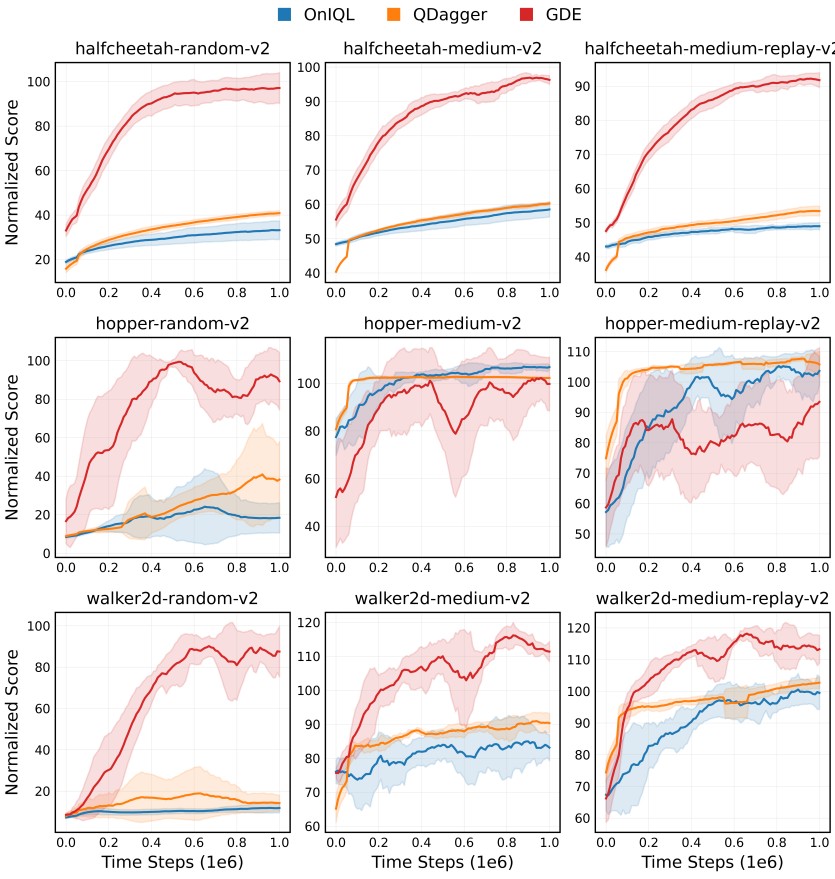

Figure 12: Fine-tuning with 1M steps on the *locomotion-v2* tasks.

## C.5   EXPERIMENTS ON ATARI GAMES

We also conducted experiments on the Atari games, where we used the CQL as the backbone algorithm. In the experiment, we first use different checkpoints of an online DQN agent trained with 10M frames to collect the offline dataset with 2M frames to train a CQL agent. We then fine-tune the offline CQL agent for another 5M frames. We compared it to the best-performing baseline, QDagger, from previous experiments. Fig 14shows the mean and standard deviation of the evaluation scores over 5 random seeds. The model is evaluated every 2.5e5 frames, and the curve is smoothed with a sliding window of 3 to facilitate visualization. We can observe that the proposed GDE outperforms QDagger in all games. A notable problem of QDagger is that it uses the fixed offline policy for policy distillation during the training, and this could be problematic when we start with a low-quality offline policy. GDE mitigates this problem by dynamically updating the teacher policy.

## D   LIMITATIONS AND FUTURE DIRECTIONS

In the Subsection 4.1 and A.3.1, we investigate two key questions in fine-tuning offline RL agents – *how to explore more effectively* and *how to leverage the offline dataset more effectively*? From the results in Table 1 and Table 5, we can observe that inefficient exploration is usually the key bottleneck and online samples are much more helpful than offline samples. Therefore, in this work, we mainly focus on addressing the *inefficient exploration* issue by introducing the *Guided Decoupled Exploration* method. For the second question of how to leverage the offline dataset, we choose a simple yet effective strategy which ignores the offline dataset completely and only uses the online samples. Though experiments show that the proposed method is effective in various tasks, ignoring the offline dataset could no way be the optimal solution. Therefore, one interesting future direction is to address this limitation and investigate how to use the offline samples without hurting the fine-

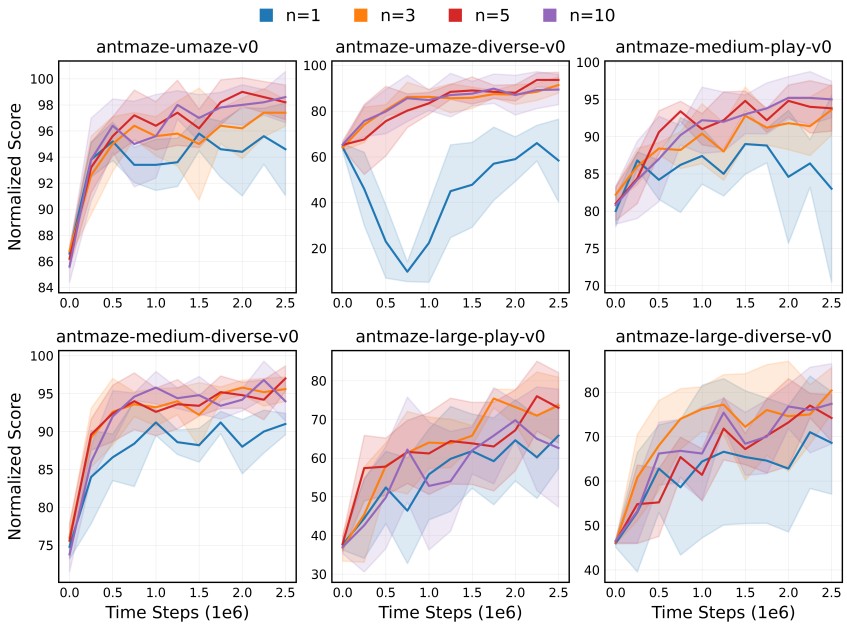

Figure 13: Ablation study with different n-step horizons on the *antmaze-v0* tasks.

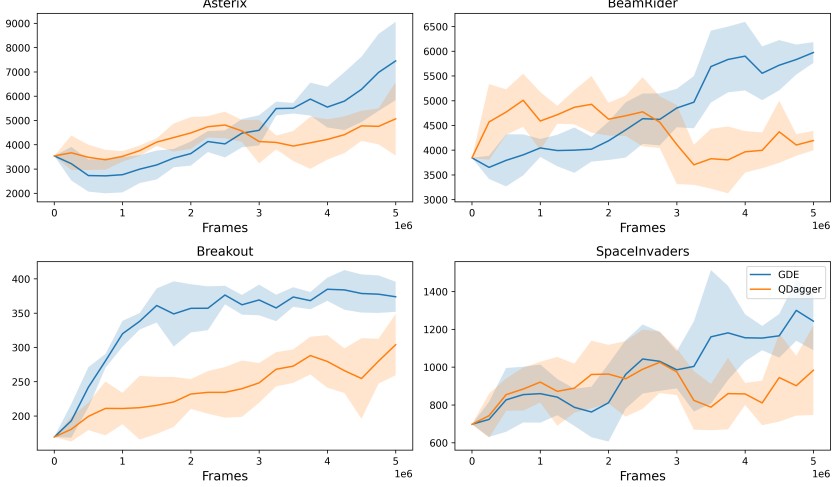

Figure 14: Experiments on four Atari games.

tuning performances. Another interesting future direction is to use meta RL to adjust different hyper-parameters automatically during the fine-tuning.

