# OpenReview forum: "Guided Decoupled Exploration for Offline Reinforcement Learning Fine-tuning"
_ICLR.cc/2024/Conference — Submitted to ICLR 2024_

### Official Review · Reviewer_XFhn · 2023-10-31

**Soundness:** 3 good
**Presentation:** 2 fair
**Contribution:** 2 fair
**Rating:** 3
**Confidence:** 4

**Summary:**

This paper identifies the excessive exploration problem that arises in an offline-to-online RL setup and proposes a new method that addresses the problem. The main idea is to separate exploration and exploitation policies and introduce a teacher policy that guides the exploration policy not to deviate too far from the teacher's actions. Teacher policy is updated to be the best policy so far by evaluating the exploitation policy at the specified regular interval. The proposed method is evaluated in locomotion and locomotion-based navigation tasks in D4RL Benchmark.

**Strengths:**

- The paper clearly motivates their paper by providing supporting analysis and experiments
- The proposed method is still simple even though it introduces several moving components. The idea of having the teacher policy that guides the exploration policy is well executed.
- The method is compared against a lot of baselines and includes the error bar, which is commendable given the current status of this field.

**Weaknesses:**

I liked reading the paper but there's some weaknesses possibly due to my understanding. Please see my weaknesses and questions.

- Introductory analysis and experiments are helpful for understanding and motivating the method but they are mostly not new as authors also already mentioned in the paper. They are mostly already covered in works like [Fujimoto'18; Lee'22; Luo'23]
- The proposed method needs evaluation rollouts for updating the teacher policy. Then the number of environment interactions required for this evaluation should also be incorporated into the sample count for the proposed method. It's not clear if this is reflected in the current results, and this should be properly computed if they are not because it's not a fair evaluation. Correctly doing this would also make Figure 6(b) analysis more meaningful because it's an important trade-off.
- The paper is a bit difficult to parse in some parts. There is a room for improving the readability. For instance, Table 4 is difficult to read because it's missing the results of GDE with all the components. Including this could help improving the readability by making not scroll the paper up and down. Figure 5 is not very helpful for understanding the main method and could be improved to intuitively help the readers to understand the main idea. Augmenting the Algorithm 1 to be more self-contained could be helpful for better readability.

[Fujimoto'18] Fujimoto, Scott, Herke Hoof, and David Meger. "Addressing function approximation error in actor-critic methods." In International conference on machine learning, pp. 1587-1596. PMLR, 2018.

[Lee'22] Lee, Seunghyun, Younggyo Seo, Kimin Lee, Pieter Abbeel, and Jinwoo Shin. "Offline-to-online reinforcement learning via balanced replay and pessimistic q-ensemble." In Conference on Robot Learning, pp. 1702-1712. PMLR, 2022.

[Luo'23] Luo, Yicheng, Jackie Kay, Edward Grefenstette, and Marc Peter Deisenroth. "Finetuning from Offline Reinforcement Learning: Challenges, Trade-offs and Practical Solutions." arXiv preprint arXiv:2303.17396 (2023).

**Questions:**

- Could you clarify what's the unique observation that could be further emphasized in the paper?
- Is the number of samples required for evaluating the policy is incorporated for counting the samples? It's important for a fair comparison.
- Is n-step used for all the methods including both the proposed method and the baselines? It's not a new component proposed in the method so it needs to be included for a fair evaluation.
- Improving some parts of Table 4, Figure 4 as in Weaknesses could be useful for improving the readability of the paper.
- In Section 5.5, it's not clear how would the method that ablates a component work. For instance, how would the method work without the exploitation policy? Then what is the main policy you are updating with? Such things are not clear so it's difficult to understand what's going on in the analysis.
- Please consider only making the numbers be bold when they are statistically significant, i.e., when errors do not overlap

---

> ### Author Response · Authors · 2023-11-23
> **Responses to Reviewer XFhn [Part 1]**
>
> We thank the reviewer for the constructive review and insightful comments. Responses to the questions are below.
>
> > **Q1: Introductory analysis and experiments are helpful for understanding and motivating the method but they are mostly not new as authors also already mentioned in the paper.**
>
> Thank you for the suggestion. Indeed, we acknowledge that certain aspects discussed in Section 3 have been covered in prior works. In Section 3, we intended to summarize and revisit some practical challenges in offline RL fine-tuning. In our view, this introductory analysis contributes to presenting a more comprehensive background and enhances the overall coherence of the paper.
>
> Contrasting our work with prior research, notable distinctions include:
>
> - [Fujimoto '18] delved into the bootstrapping error issue within offline RL training.
> - [Lee '22] focused on addressing the bootstrapping error during the online fine-tuning stage.
> - [Luo '23] explored the policy crash issue during the online fine-tuning stage.
>
> In this work, we synthesized these distinct elements and introduced a straightforward algorithm to tackle these challenges. Unlike [Fujimoto '18, Lee '22, Luo '23], which proposed specific algorithms in their respective papers, we presented a general framework. The experiments illustrate its effectiveness in collaboration with various backbone offline RL agents.
>
>
>
> > **Q2: Readbilities can be improved**
>
> Appreciate the suggestions. We have revised Table 4, Figure 5, and Algorithm 1 to enhance readability.
>
>
>
> > **Q3: unique observations**
>
> We acknowledge that some observations in our paper overlap with prior works. However, we believe certain unique observations can be highlighted:
>
> - The optimal way to leverage offline samples during fine-tuning remains an open question. In some cases, dropping the offline buffer proves to be a surprisingly strong baseline.
> - It might be helpful to use varying degrees of conservatism in exploration and training, as demonstrated by the decoupling method.
>
>
>
> > **Q4: Is the number of samples required for evaluating the policy is incorporated for counting the samples?**
>
> The evaluation trials were not considered in the budget for online fine-tuning. Consequently, we made an error by utilizing additional online samples for policy selection.
>
> As discussed in the common question section, we have rectified this issue. The overall results align closely with the previous findings. The results in Table 2, Table 3, Table 4, and Figure 6 are updated accordingly.

---

> > ### Author Response · Authors · 2023-11-23
> > **Responses to Reviewer XFhn [Part 2]**
> >
> > > **Q5: is n-step return used for the baselines**
> >
> > Thank you for the reminder. In our previous experiments, we didn't utilize the n-step return for the baselines, and we acknowledge that this could lead to an unfair comparison. To address this concern, we have compared GDE to the two best-performing baselines using n-step returns. We can observe that GDE continues to outperform QDagger and PEX.
> >
> > |             |  QDagger   |    PEX     |    GDE     |
> > | :---------: | :--------: | :--------: | :--------: |
> > |  antmaze-u  | 96.6 (1.0) | 97.4 (1.0) | 97.6 (1.0) |
> > | antmaze-u-d | 85.4 (2.9) | 90.2 (1.6) | 90.2 (2.5) |
> > | antmaze-m-p | 90.6 (2.4) | 92.0 (2.3) | 94.8 (2.1) |
> > | antmaze-m-d | 91.4 (2.1) | 94.8 (1.7) | 94.4 (2.1) |
> > | antmaze-l-p | 73.0 (3.6) | 71.6 (2.9) | 73.2 (6.7) |
> > | antmaze-l-d | 67.8 (4.9) | 69.6 (4.0) | 80.4 (2.3) |
> > |    Total    |   504.8    |   515.6    |   530.6    |
> >
> >
> > > **Q6: readability of Table 4**
> >
> > Thank you for the suggestions. We have revised Table 4 to highlight the worst-case performance.
> >
> >
> > > **Q7:  it's not clear how would the method that ablates a component work**
> >
> > Thank you for the suggestions. We have revised Section 5.5 for clarity.
> >
> > In the ablation study on not using an exploitation policy, we experimented with two policies, utilizing the checkpoint of the exploration policy as the teacher policy. The exploration policy is updated according to Eqn (5).
> >
> > > **Q6: only bold number when they are statistically significant**
> >
> > Thank you for the suggestion. We have revised the tables in the paper accordingly.

---

> > > ### Comment · Reviewer_XFhn · 2023-11-23
> > >
> > > Thank you for your response and additional results. Currently I have no questions or concerns, but there could be some things that I missed as I quickly went through the responses because there is not many time left. I'll go through the other reviews and responses and update the score during the internal discussion period.

---

### Official Review · Reviewer_uJfn · 2023-11-01

**Soundness:** 3 good
**Presentation:** 2 fair
**Contribution:** 2 fair
**Rating:** 5
**Confidence:** 4

**Summary:**

The paper presents three issues afflicting the performance of offline-to-online fine-tuning methods: insufficient exploration due to conservative pre-training, distribution-shift between offline and online distribution, distorted value functions. The paper then motivates and designs an algorithm based on IQL, where a decoupled exploration policy collects the online data. The loss for the exploration is based on the TD3 update, in contrast with the AWR loss used for policy update in IQL. However, TD3 update can cause the policy performance to crash, to avoid which a KL penalty with the best exploitation policy (aka teacher policy) is introduced, which also helps take safer actions. Overall, the proposed framework allows for more efficient offline-to-online fine-tuning.

**Strengths:**

- Good coverage of the literature
- The discretization into the three problems for offline-to-online RL makes sense and pedagogically useful
- Section 3.2 presents an interesting experiment where removing the offline data during online fine-tuning improves the performance. Presenting more concrete evidence for this observation would improve the paper further.
- Lots of experiments and the performance improvements are substantial, though more comparisons are needed to ascertain if the gains are for the hypothesized reasons.

**Weaknesses:**

The clarity of Section 4 and method description can be improved quite a bit. Please see questions for further clarifications.

Overall, I am not entirely sure about the generality of the framework, while the paper claims to be general. For example, if the actor loss for the base algorithm already uses a TD3 update, it is not obvious to me that the decoupled exploration does anything different.

The main benefit of GDE likely comes from the ability to allow the behavior policy to be updated using TD3, which cannot be done with IQL naively, as Q-values aren’t  trained on OOD actions in IQL. The policy crash at low levels of $\rho$ is likely because of using TD3 loss with an IQL trained Q-value function. This make the comparison with [2] quite critical. They report fairly sample efficient results, but beyond empirical gains, I suspect most of the benefit in GDE is derived from the fact that using a TD3 update for exploration policy, which improves the policy much faster than AWR update and thus generates better exploration data. CalQL shows that calibrated Q-value functions can allow direct usage of TD3 style updates for policy improvement, without the whole decoupled framework.

Overall I am willing to improve the score for the paper, if some of these conditions can be met:

(1) The phenomenon in 3.2 is established in more environments, with algorithms beyond IQL

(2) Comparisons with CalQL are added, and GDE outperforms CalQL.

(3) Alternately, the framework is shown to be compatible with CalQL, and demonstrates an improvement in the performance over it

(4) Clarifying the writing in Section 4, and confirming that the evaluation rollouts for exploitation policy are duly counted in the fine-tuning budget


[1] Beyond Uniform Sampling: Offline Reinforcement Learning with Imbalanced Datasets. Hong et al. NeurIPS 2023.

[2] Cal-QL: Calibrated Offline RL Pre-Training for Efficient Online Fine-Tuning. Nakamoto et al. NeurIPS 2023.

**Questions:**

- Section 3.1 inefficient exploration; have you tried adding RND reward to the offline agent to encourage wider/broader exploration? This issue seems to be part of the desiderata
- Section 3.2: This is an interesting point, but might be worth rephrasing the open question  “how to leverage prior knowledge without hurting performance” to clarify how to best use offline data during online fine-tuning. Figure 3 (a) is quite interesting. Am I understanding it correctly that for IQL and SAC, the policy and Q-values are pre-trained (using IQL and CQL respectively on the offline data), but removing the offline data during online fine-tuning and collecting data in the replay buffer from scratch improves the performance and efficiency during training? Can you reproduce this phenomenon on other environments, potentially AntMaze or Kitchen environments? Would this be less of a problem if the offline data contained higher proportion of expert trajectories? One possible way to continue using offline data during online fine-tuning is to rebalance the offline distribution, sampling more relevant transitions more frequently. See [1].

For Table 1:
- Can you report the default IQL performance for comparison, ie, IQL that continues to use offline data naively during online fine-tuning? If using the official code for IQL, please note that it is not setup to fine-tune on locomotion environments, as the reward using during offline and online fine-tuning are different. Fixing that is important before reporting the IQL fine-tuning results.
- Can you report the average performances as well (clustering locomotion envs, antmaze environments)?

Section 4:
- The clarity of writing in this section can be improved as it is missing quite a few details — more explicit details would greatly improve the understanding of GDE + minimizing or defining notation clearly with \phi, \mu, \hat{\phi}, \bar{\phi}
- Is exploration policy initialized randomly?
- How many trials are done for evaluation? Are those trials counted towards the budget for online fine-tuning? Are you evaluating every 2500 (Appendix B3) steps or 25000 steps (Table 7)?
- How is the exploitation policy updated over the course of training? What does “exploitation policy pi_e which is responsible for policy extraction from newly collected online samples mean”? Is it updated only on the online samples or does it continue to use offline samples?
- Do the exploration policy and exploitation update on the same value functions? I understand that this is possible to do for IQL where the Q-values are trained using the replay buffer data and does not query the policy, but if you use a different base algorithm, for example, CalQL, would this still make sense?
- Which policy is used for reporting the performance? Have you tried evaluating the exploration policy?

---

> ### Author Response · Authors · 2023-11-23
> **Responses to Reviewer uJfn [Part 1]**
>
> We thank the reviewer for the constructive review and insightful comments. Responses to the questions are below.
>
> > **Q1: The clarity of Section 4 and method description can be improved quite a bit**
>
> Thank you for the suggestion. We have rewritten Section 4 and the method description to enhance clarity.
>
>
>
> > **Q2: not entirely sure about the generality of the framework, while the paper claims to be general. For example, if the actor loss for the base algorithm already uses a TD3 update, it is not obvious to me that the decoupled exploration does anything different.**
>
> In stating, 'It is notable that the proposed GDE is a general method that can be combined with different offline RL algorithms,' our intention was to convey that GDE can be employed to fine-tune various offline RL agents.
>
> Unlike certain other approaches, such as CAL-QL, which is a specific offline RL fine-tuning algorithm limited to a single backbone offline RL algorithm, GDE is a framework indifferent to the choice of backbone algorithms.
>
> Indeed, our method aligns with TD3+BC in using the same TD3 actor loss. The distinction lies in GDE introducing adaptive conservatism for the exploration and exploitation policy. Here, we encourage the exploration policy to take dissimilar actions while maintaining that the exploitation policy remains more conservative to avoid the policy crash issue.
>
>
>
> > **Q3: I suspect most of the benefit in GDE is derived from the fact that using a TD3 update for exploration policy,**
>
> Excellent question. We concur with the reviewer that the primary advantages of GDE stem from improved exploration data. As highlighted in the concluding section of the introduction, 'We then focus on addressing the exploration issue to achieve sample-efficient fine-tuning,' our emphasis is on mitigating the inefficient exploration problem arising from an overly conservative policy in offline RL fine-tuning.
>
> In this paper, utilizing a TD3-like update for the exploration policy serves as a straightforward method to validate our assumptions. We acknowledge that the TD3-like update is a common factor contributing to the effectiveness of both GDE and CAL-QL.
>
> We introduced the decoupled teacher policy to prevent policy crash issues, whereas CAL-QL addresses this matter through a modified CQL critic loss. GDE and CAL-QL aim to resolve this issue using different methods.
>
> The key distinction lies in our decoupled framework's ease of integration with other offline RL algorithms.
>
>
>
> > **Q4: Phenomenon in 3.2 with on more environments with more algorithms. Will more expert data help?**
>
> Thank you for the feedback. We have incorporated the results of fine-tuning a TD3+BC agent with and without the offline buffer into Figure 3(a), and the outcomes align with those of IQL and SAC.
>
> As mentioned in our response to Q1 for R1, we have conducted similar experiments where we trained a SAC agent from scratch using a preload buffer. Our observations indicate that omitting the offline replay buffer proves beneficial when the quality of offline data is low, such as in the case of a random dataset.
>
> To further investigate this phenomenon across multiple environments, we fine-tuned a TD3+BC agent and a SAC agent using the pretrained checkpoint of a TD3+BC and a CQL offline RL agent, respectively. The average evaluation scores at the end of 2.5e5 steps are reported below:
>
> |                  | TD3+BC without buffer | TD3+BC with buffer | SAC without buffer | SAC with buffer |
> | :--------------: | :-------------------: | :----------------: | ------------------ | --------------- |
> |  halfcheetah-r   |      49.1 (2.3)       |     35.5 (1.6)     | 80.7 (2.4)         | 68.7 (1.6)      |
> |     hopper-r     |      17.1 (3.4)       |     9.9 (0.3)      | 102.4 (10.8)       | 43.8 (21.8)     |
> |    walker2d-r    |       7.0 (2.6)       |     5.1 (5.7)      | 39.1 (14.6)        | 15.9 (6.8)      |
> |  halfcheetah-m   |      71.0 (1.5)       |     56.6 (0.5)     | 85.6 (1.8)         | 66.6 (0.9)      |
> |     hopper-m     |      102.3 (0.3)      |     60.1 (3.2)     | 100.4 (12.9)       | 105.7 (0.2)     |
> |    walker2d-m    |      70.7 (40.5)      |     86.0 (0.7)     | 72.1 (24.3)        | 97.6 (0.3)      |
> | halfcheetah-m-re |      64.4 (3.0)       |     51.7 (0.4)     | 78.9 (7.1)         | 71.3 (0.3)      |
> |   hopper-m-re    |      103.9 (1.2)      |    79.2 (29.7)     | 108.2 (0.9)        | 102.8 (3.8)     |
> |  walker2d-m-re   |      100.4 (3.5)      |     91.1 (0.3)     | 109.0 (7.3)        | 99.0 (1.3)      |
> |      Total       |         585.9         |       475.2        | 776                | 671.4           |
>
> We note that fine-tuning a TD3+BC or SAC agent without the replay buffer tends to yield strong performance on locomotion tasks. However, these methods often struggle in antmaze tasks, resulting in 0 rewards, aligning with the observations discussed in the response to Q1 for R1.

---

> > ### Author Response · Authors · 2023-11-23
> > **Responses to Reviewer uJfn [Part 2]**
> >
> > > **Q5: Compare to CAL-QL**
> >
> > Thanks for the suggestion! We answered this question in the common question section.
> >
> >
> >
> > > **Q6: RND reward**
> >
> > Thank you for your suggestion. We have included the results of using RND in Table 1, where RND proves to be beneficial in some tasks. While further parameter tuning for RND might improve results, it's essential to note that merely adding an exploration intrinsic reward doesn't address the underlying issues in the offline RL agent's policy loss.
> >
> >
> >
> > > **Q7: reproduce phenomenon on Antmaze task**
> >
> > Thank you for your suggestion. We have revised the related section for improved clarity.
> >
> > Regarding the training details, our method was implemented exactly as described.
> >
> > Concerning the observation of IQL without the offline buffer on the Antmaze task, we did not encounter this phenomenon. The antmaze task, being a reward-sparse environment, can lead to the policy crash issue during fine-tuning. Our approach mitigates this problem by introducing a teacher policy, and the preload offline dataset can implicitly act as a teacher, given that IQL is a variant of the behavior cloning algorithm.
> >
> > The observed phenomenon is closely tied to the trade-off between fast learning and safe learning. For instance, in locomotion tasks, prioritizing fast learning without excessive concern for safety could make dropping the offline buffer beneficial. Conversely, in the antmaze task, where safe learning is crucial, dropping the offline buffer may not yield favorable results.
> >
> > Our method aims to strike a balance between fast and safe learning by combining the strategies of dropping the offline dataset and utilizing a teacher policy.
> >
> > |             | IQL with Offline Buffer | IQL without Offline Buffer | GDE        |
> > | :---------: | :---------------------: | :------------------------: | ---------- |
> > | antmaze-l-p |       49.2 (3.8)        |        24.5 (20.4)         | 73.2 (6.7) |
> > | antmaze-l-d |       49.4 (6.2)        |        11.7 (10.3)         | 80.4 (2.3) |
> >
> >
> >
> > > **Q8: Would this be less of a problem if the offline data contained higher proportion of expert trajectories? One possible way to continue using offline data during online fine-tuning is to rebalance the offline distribution, sampling more relevant transitions more frequently.?**
> >
> > Honestly, I am not entirely certain about the answer to this question at this juncture. In my opinion, it could be a promising starting point for an individual research project.
> >
> > Undoubtedly, offline data is pivotal in certain tasks, and simply discarding them may not be the optimal choice. It is likely that the impact of offline data would be less problematic if the dataset contains a higher proportion of expert trajectories. Another promising solution could be to sample more relevant transitions more frequently.
> >
> > However, addressing this question fundamentally appears to be challenging. Intuitively, the answer might depend on various factors such as:
> >
> > - The difficulty of the task: Is it a reward-sparse or reward-dense task?
> > - The diversity of the offline dataset: Does it predominantly consist of expert data or lower-quality data?
> > - The disparity between the behavior policy in the offline dataset and the current policy: Could off-policy data lead to unstable optimization?
> > - The strategy for handling the offline data during fine-tuning: Should all the data be retained, or is there a need for selective sampling?
> >
> > In our work, our intention is to highlight the open question of how to leverage offline datasets more effectively, a aspect that seems to have been overlooked in some prior work.
> >
> > We eagerly anticipate future research endeavors aimed at addressing this challenging problem.

---

> > > ### Author Response · Authors · 2023-11-23
> > > **Responses to Reviewer uJfn [Part 3]**
> > >
> > > > **Q9: default IQL in Table 1**
> > >
> > > Thanks for the suggestion. The IQL result in Table 1 actually follows the default fine-tuning strategy where we add online samples to the offline buffer. Additionally, we used our reproduced IQL in the experiment, and the reward shaping is consistent across the offline-training and online fine-tuning stages.
> > >
> > >
> > >
> > > > **Q10: average result in Table 1**
> > >
> > > Thanks for the suggestion; we updated the average performances in Table 1.
> > >
> > >
> > >
> > > > **Q11: is exploration policy initialized randomly?**
> > >
> > > No, all three policies are initialized using the pre-trained offline RL policy.
> > >
> > >
> > >
> > > > **Q12: evaluation details**
> > >
> > > We mainly followed the experiment setups from some prior work, evaluating the agent in the locomotion and antmaze tasks for 10 and 100 trajectories, respectively.
> > >
> > > The evaluation trials were not counted in the budget for online fine-tuning. Therefore, we made a mistake by using extra online samples to select the policy. We have corrected this issue, as discussed in the common question section. The overall results are similar to the previous ones.
> > >
> > > We evaluate every 2500 steps for the locomotion tasks and every 25000 steps for the antmaze tasks. This is mainly because we followed some prior work to plot training curves with 100 points and 10 points for the locomotion and antmaze tasks, respectively.
> > >
> > >
> > >
> > > > **Q13: how is exploitation policy updated**
> > >
> > > We followed the actor loss of the backbone offline RL algorithm to update the exploitation policy.
> > >
> > >
> > >
> > > > **Q14: Do the exploration policy and exploitation policy update on the same value functions?**
> > >
> > > Yes, we use the same value function to update both the exploration policy and exploitation policy.
> > >
> > > While it's possible to maintain two separate value functions for these policies, we have found that this simpler approach works effectively in practice. For instance, in Table 3, we utilize this method to train a GDE agent based on TD3+BC and CQL.
> > >
> > >
> > > > **Q15: which policy is used to report the performance**
> > >
> > > We present the performance metrics for the exploitation policy. Additionally, we examined the results of the exploration policy, which showed slightly lower scores than the exploitation policy and exhibited higher variance.

---

### Official Review · Reviewer_Se55 · 2023-11-04

**Soundness:** 3 good
**Presentation:** 4 excellent
**Contribution:** 3 good
**Rating:** 6
**Confidence:** 3

**Summary:**

This paper investigates the challenges of offline-to-online RL with interesting experimental exploration, namely inefficient exploration, distributional shifted samples and distorted value functions. Based on the empirical findings and analysis, it proposes a simple yet effective algorithm called Guided Decoupled Exploration (GDE), which maintains a exploration policy and a teacher policy in addition to the main exploitation policy. GDE outperforms prior approaches in multiple domains with various backbone algorithms. Ablation study and hyperparameter tests are provided to verify the effectiveness of GDE.

**Strengths:**

- This paper studies an interesting and important problem, which is to finetune an offline learned policy in online environments.
- The paper starts by demonstrating the key challenges in this setup with motivating experiments. Although the studied challenges have been discussed a lot in literature, the experiments in Sec 3 provide factual evidence, which I find interesting and helpful.
- The proposed algorithm is based on the empirical findings, which makes intuitive sense and works well in standard benchmarks.

**Weaknesses:**

- The analysis of challenges can be made more in-depth. The current experiments are more like proof-of-concept and the results can be expected.
- GDE maintains 3 policies, rendering extra computation and memory costs. Although the authors emphasize the minimalist algorithm design, I feel that the current design is not necessarily the most efficient. For example, can the exploration poilcy directly be a function of the exploitation policy (one can just adjust the output action distribution by exploration objectives, without training an extra policy.)

**Questions:**

1. There are a lot of exploration approachs, is there a reason of selecting Eq (5) as the loss for the exploration policy? Will other methods work here, such as curiosity-driven ones?
2. Why is the performce of train from scratch with SAC so low? Given enough samples, shouldn't SAC be able to learn a good policy in many of these tasks?

---

> ### Author Response · Authors · 2023-11-23
> **Responses to Reviewer Se55**
>
> We thank the reviewer for the constructive review and insightful comments. Responses to the questions are below.
>
> > **Q1: The analysis of challenges can be made more in-depth.**
>
> Thank you for the valuable suggestion. We acknowledge that our focus has been primarily on illustrating the challenges in current offline RL fine-tuning. We provided analysis on these challenges in Appendix B.5.
>
> Given the constraints of space, delving deeply into the three challenges might be challenging. Our approach in this work has been to analyze key observations and demonstrate the efficacy of the proposed method. We are open to the prospect of conducting a more in-depth theoretical analysis of these challenges in future work.
>
>
> > **Q2: GDE maintains 3 policies, rendering extra computation and memory costs.**
>
> Yes, GDE indeed maintains three policies: a teacher policy, an exploration policy, and an exploitation policy. It's important to note that the teacher policy is essentially a checkpoint of the previous exploitation policy. Consequently, during training, we only need to train two policies. Moreover, the exploration policy is implemented as a simple 2-layer MLP, which means the additional computation overhead is relatively light.
>
>
> > **Q3:  can the exploration policy directly be a function of the exploitation policy**
>
> Excellent question. The primary focus of this paper is to outline practical challenges in current offline RL fine-tuning, specifically inefficient exploration due to the over-conservative policy, and then propose a straightforward solution to address this issue. The current design was chosen for its simplicity and intuitiveness.
>
> Acknowledging that there might be more efficient algorithm designs, one potential solution to make the exploration policy directly a function of the exploitation policy is to modify the current policy structure into a conditioned policy $\pi(a\vert s, z)$. Here, we would learn a new latent variable $z$ to control conservatism during action sampling, thereby unifying the exploration policy and exploitation policy. The exploration of effective training methods for such a policy is a topic we leave for future work.
>
>
>
> > **Q4: Why select Eq(5)? Will curiosity-driven method work?**
>
> One of the main reasons for using Equation 5 lies in its simplicity. Beginning with a minimalist perspective in designing our algorithm, we opted to add a single regularization term to the canonical actor loss in the TD3 and SAC.
>
> Our primary goal was to demonstrate that by introducing this new actor loss, we could address the issue of inefficient exploration while staying relatively close to the behavior policy. We believe that other actor losses with a similar concept could yield positive results as well, and we chose one of the simplest forms to validate our assumptions. For instance, incorporating a Random Network Distillation (RND) module to augment the task reward and encourage exploration could be beneficial in certain tasks, as illustrated in Table 1.
>
> However, such methods typically entail learning an additional model and introduce additional computational complexity. Given the effective performance of our current simple design in experiments, we opted to retain it.
>
>
> > **Q5: Performance of Fromscratch SAC**
>
> The seemingly low performance of 'Fromscratch SAC' can be attributed to two primary reasons:
>
> - **Limited Interaction Steps:** Our focus in this study was on the challenge of sample-efficient offline RL fine-tuning. Consequently, we constrained the agent to interact with the environment for only 2.5e5 environmental steps in the experiments. The suboptimal performance of Fromscratch SAC suggests that the randomly initialized policy is less efficient in collecting useful samples during the early stages of learning.
> - **Normalized Evaluation Scores:** The reported scores are normalized evaluation scores, and as a result, they may appear low. If we were to run Fromscratch SAC for 1e6 environmental steps, it would likely achieve scores near 100 in the locomotion tasks.

---

### Official Review · Reviewer_DsmU · 2023-11-05

**Soundness:** 3 good
**Presentation:** 3 good
**Contribution:** 3 good
**Rating:** 8
**Confidence:** 4

**Summary:**

The paper describe a novel technique were exploration in offline to online RL is decoupled from the offline learning. This guided exploration avoid three of the current pitfalls of the offline RL techniques. Inefficient exploration because of biased conservatism. The difference in probability distribution between offline and online samples. Finally the value function learned from the offline dataset is far away from the optimal value function. In this work a teacher policy is introduced which guide the exploration policy to avoid policy crashing. The teacher policy is updated frequently. This decoupled avoid the conservatism bias and focusing on the latest online samples and using a n-step return made this algorithm more sample efficient.

**Strengths:**

originality: It's quite novel the approximation even though the community have proposed related solution for some of the problems. CA-CQL for efficiency when jumping to the online phase. (Some concurrent work that also use decoupling: Offline Retraining for Online RL: Decoupled Policy Learning to Mitigate Exploration Bias Mark et al 2023) What I like about the paper is that address all the three approach in one coherent algorithm.

 quality: the paper have presented clear equation to backup the claims, and have provided a strong methodology and well written experiments section, with proper problems accepted by the community.

clarity: the paper is quite clear in its presentation, the structure and flow of the paper is well done.

 significance: the paper in an incremental change on the field of offline RL.

**Weaknesses:**

Probably one weakness I see is how it compared with a off-policy algorithm with a preload buffer.
It would interesting to see how it compared with CA-CQL as well.

**Questions:**

What's not clear to me in this paper is what the difference between this and having a off-policy algorithm  that start with a preload buffer?

---

> ### Author Response · Authors · 2023-11-13
> **A question about the CA-CQL baseline.**
>
> Dear Reviewer,
>
> Could you kindly tell me which paper is the CA-CQL method or if you intended to mention Cal-QL [1]?
>
> [1] Cal-QL: Calibrated Offline RL Pre-Training for Efficient Online Fine-Tuning

---

> > ### Comment · Reviewer_DsmU · 2023-11-13
> >
> > Cal-QL

---

> > > ### Author Response · Authors · 2023-11-13
> > >
> > > Thank you. We will report the results for the Cal-QL baseline later.

---

> ### Author Response · Authors · 2023-11-23
> **Responses to Reviewer DsmU**
>
> We thank the reviewer for the constructive review and insightful comments. Responses to the questions are below.
>
> >  **Q1: how it compared with an off-policy algorithm with a preload buffer.**
>
> Thanks for the question. We have incorporated experiments in Appendix C.1 to investigate the performance of training an off-policy algorithm with a preload buffer. We compare GDE to training a SAC agent from scratch with and without a preload buffer, respectively.
>
> |                  | SAC without buffer | SAC with buffer |       GDE       |
> | :--------------: | :----------------: | :-------------: | :-------------: |
> |  halfcheetah-r   |   **59.5 (5.4)**   |   52.4 (2.4)    |   57.6 (13.4)   |
> |     hopper-r     |    82.7 (19.1)     |   25.6 (4.0)    | **92.6 (7.0)**  |
> |    walker2d-r    |  **47.4 (17.8)**   |   15.4 (8.1)    |   33.3 (20.3)   |
> |  halfcheetah-m   |     59.5 (5.4)     |   61.1 (0.8)    | **75.7 (1.8)**  |
> |     hopper-m     |    82.7 (19.1)     |   66.6 (11.2)   | **101.9 (6.9)** |
> |    walker2d-m    |    47.7 (17.8)     |   73.0 (11.2)   | **101.7 (6.1)** |
> | halfcheetah-m-re |     59.5 (5.4)     |   67.5 (1.8)    | **70.9 (1.2)**  |
> |   hopper-m-re    |    82.7 (19.1)     |   85.1 (19.4)   | **101.5 (8.0)** |
> |  walker2d-m-re   |    47.7 (17.8)     |   84.0 (1.1)    | **100.9 (3.9)** |
> |      Total       |       569.7        |      530.7      |    **736.1**    |
>
> |             | SAC without buffer | SAC with buffer |      GDE       |
> | :---------: | :----------------: | :-------------: | :------------: |
> |  antmaze-u  |       0 (0)        |    2.4 (0.5)    | **97.6 (1.0)** |
> | antmaze-u-d |       0 (0)        |      0 (0)      | **90.2 (2.5)** |
> | antmaze-m-p |       0 (0)        |      0 (0)      | **94.8 (2.1)** |
> | antmaze-m-d |       0 (0)        |      0 (0)      | **94.4 (2.1)** |
> | antmaze-l-p |       0 (0)        |      0 (0)      | **73.2 (6.7)** |
> | antmaze-l-d |       0 (0)        |      0 (0)      | **80.4 (2.3)** |
> |    Total    |         0          |       2.4       |   **530.6**    |
>
> We can observe that training a SAC agent with a preload buffer yields satisfactory results on the locomotion tasks. However, its performance significantly degrades on more challenging antmaze tasks.
>
> The primary reason behind this discrepancy lies in the nature of the antmaze task, characterized by sparse rewards and goal-reaching challenges. The preload buffer contains sub-optimal trajectories that make direct training of an SAC agent challenging.
>
> Usually, we need to leverage the *stitching ability* from offline RL to learn a sub-optimal policy, and then fine-tune the policy with further online interactions.
>
>
>
> > **Q2: comparison with CAL-QL**
>
> Appreciate the suggestion! We have addressed this query in the Common Questions section.
>
>
>
> > **Q3: difference w.r.t. an off-policy algorithm with a preload buffer**
>
> Thank you for the thoughtful question. Our method differs from an off-policy algorithm with a preload buffer in the following key aspects:
>
> - **Knowledge Transfer:**
>     - Our method involves fine-tuning a pre-trained offline RL checkpoint with additional online interactions to enhance performance. This means transferring the learned policy and value function from the offline learning stage to the online learning stage.
>     - In contrast, training an off-policy algorithm with a preload buffer primarily transfers collected samples from the offline learning stage to the online learning stage.
> - **Curriculum Learning Perspective:**
>     - While the off-policy algorithm with a preload buffer is a general approach applicable to all tasks, it may face challenges in quickly learning certain challenging tasks, such as the antmaze scenarios. The preload samples might not be effective enough to initiate efficient learning.
>     - Our proposed method, on the other hand, employs a pre-trained policy to collect online samples during the fine-tuning stage, providing the agent with a more manageable task.
>     - This difference can be interpreted through the lens of curriculum learning. The off-policy algorithm with a preload buffer might expose the agent to a challenging task right from the start, whereas our method, using the pre-trained policy, allows the agent to first collect sub-optimal trajectories, easing it into the learning process."

---

### Author Response · Authors · 2023-11-23
**Common Questions**

> **Q1:  use more online samples**

Thanks for the reviewers for raising this point.

We admit that the current model will use extra online samples to select teacher models, which leads to an unfair comparison to other baselines. To address this problem, we now explicitly divide the training process of GDE into two iterative phases -- (1) the online fine-tuning phase and (2) online policy-evaluation phase.  If we use [====] to represent the online fine-tuning phase and use [x] to represent the online evaluation phase. Then the training process of GDE looks like [====] [x] [====] [x] ...... [====].

In the corrected experiments, we make sure that the total environment steps of [====] and [x] is the same as the other baselines. We further evaluate GDE on multiple short trajectories in [x] instead of on full trajectories to save more online interactions for [====]. Moreover, samples collected in [x] are added to the replay buffer to train the agent. In the revised paper, we rewrote the Section 4.2.3 to clarify this issue.

Some updated experiment results are summarized as follows, where the corrected results generally align closely with the previous findings:




|                  |  GDE 250K   |   Updated   |   GDE 1M    |   Update    |
| :--------------: | :---------: | :---------: | :---------: | :---------: |
|  halfcheetah-r   | 58.4 (20.6) | 57.6 (13.4) | 96.8 (6.7)  | 95.3 (6.8)  |
|     hopper-r     | 91.8 (8.0)  | 92.6 (7.0)  | 92.8 (13.9) | 96.1 (13.7) |
|    walker2d-r    | 28.5 (16.9) | 33.3 (20.3) | 86.3 (9.9)  | 86.0 (11.0) |
|  halfcheetah-m   | 86.0 (2.4)  | 75.7 (1.8)  | 96.8 (0.8)  | 96.8 (0.8)  |
|     hopper-m     | 98.6 (19.4) | 101.9 (6.9) | 102.2 (8.9) | 104.2 (8.8) |
|    walker2d-m    | 101.5 (6.3) | 101.7 (6.1) | 112.2 (1.8) | 111.4 (1.2) |
| halfcheetah-m-re | 73.5 (2.5)  | 70.9 (1.2)  | 92.2 (1.6)  | 92.2 (1.6)  |
|   hopper-m-re    | 93.9 (15.2) | 101.5 (8.0) | 91.9 (16.9) | 98.6 (11.5) |
|  walker2d-m-re   | 102.5 (5.2) | 100.9 (3.9) | 114.1 (4.2) | 114.1 (4.7) |
|    antmaze-u     | 97.4 (1.0)  | 97.6 (1.0)  | 98.2 (2.2)  | 98.2 (2.2)  |
|   antmaze-u-d    | 91.4 (3.6)  | 90.2 (2.5)  | 94.4 (7.8)  | 98.2 (1.5)  |
|   antmaze-m-p    | 93.6 (3.4)  | 94.8 (2.1)  | 97.0 (1.7)  | 97.0 (1.9)  |
|   antmaze-m-d    | 95.6 (1.6)  | 94.4 (2.1)  | 99.2 (0.4)  | 99.2 (0.4)  |
|   antmaze-l-p    | 73.6 (7.3)  | 73.2 (6.7)  | 91.4 (1.9)  | 91.5 (2.1)  |
|   antmaze-l-d    | 80.4 (5.2)  | 80.4 (2.3)  | 86.8 (4.9)  | 88.0 (5.0)  |




> **Q2: compare to CAL-QL**

Thanks for the suggestion! We have incorporated additional experiments comparing GDE to CAL-QL in Appendix C.2.

It's important to clarify some nuanced differences between GDE and CAL-QL:

- CAL-QL is a specific two-stage offline RL fine-tuning algorithm, whereas GDE is a general offline RL fine-tuning framework.
    - CAL-QL initially employs a modified CQL loss to train an offline RL agent, followed by fine-tuning this agent using a mixture of offline and online samples.
    - GDE, on the other hand, is agnostic to the offline learning stage and can be paired with any other backbone offline RL agents.

- Consequently, we cannot directly integrate CAL-QL results into Table 2 or Table 3, where we use IQL/TD3+BC/CQL as the pretrained offline checkpoint for fine-tuning instead of CAL-QL.

- We first use the official code (https://github.com/nakamotoo/Cal-QL) to train the CAL-QL offline agent, and then use GDE to fine-tuning the trained CAL-QL agent for 2.5e5 steps as in Table 2 and Table 3.

 We can observe that GDE is also effective when we use a CAL-QL based backbone algorithm.

|        | antmaze-m-p | antmaze-m-d | antmaze-l-p | antmaze-l-d |
| ------ | ----------- | ----------- | ----------- | ----------- |
| CAL-QL | 89.6 (0.5)  | 91.0 (1.7)  | 75.2 (2.6)  | 79.6 (1.9)  |
| GDE    | 91.0 (1.1)  | 92.4 (1.4)  | 77.4 (2.6)  | 80.0 (2.5)  |

---

### Author Response · Authors · 2023-11-23
**General reply**

We express our gratitude to all the reviewers for dedicating their time and effort to provide valuable feedback on our paper. In response to the reviewers' comments, we have carefully addressed and incorporated the suggested improvements into the revised manuscript. We will first provide a response summary here and later address each reviewer's concerns in separate responses. In the following replies, we will use R1/R2/R3/R4 to refer to the reviewer DsmU, Se55, uJfn and XFhn, respectively.



We are happy to see that reviewers generally agree that: (1) this work is well motivated [R1, R2, R4]; (2) the presentation is clear [R2, R3, R4]; and (3) the experiment results are promising [R1, R2, R3, R4].



Following the insightful feedback from the reviewers, we have:

- We have corrected the inadvertent use of extra online samples for teacher policy selection and updated the experimental results accordingly. [R3, R4]
- Add comparison to CAL-QL. [R1, R3, R4]
- Add experiments for off-policy agent + preload buffer. [R1]
- Update Figure 3(a) to add results of TD3+BC. [R3]
- Add experiments to finetune TD3+BC and SAC agents with and without offline buffer. [R3]
- Add experiments to finetune IQL without buffer on the Antmaze tasks. [R3]
- Rewrote Section 4 and experiment description to improve the clarity. [R3, R4]
- Add experiments for the baselines to use nstep returns. [R4]
- Update the tables, Algorithm 1 and Figure 5 to improve readability. [R4]

---

### Comment · Area_Chair_coTg · 2023-11-23
**Author-Reviewer discussion period ending *very* soon**

Dear reviewers (XFhn, uJfn,Se55,DsmU). The authors have put great effort into their response, so can I please urge you to answer the rebuttal.
Thank you!

---

### Meta-Review · Area_Chair_coTg · 2023-12-06

**Metareview:**

The paper addresses the sample-efficient fine-tuning of offline-learned policies in online environments, tackling three significant challenges: inefficient exploration, distributional shifts in samples, and distorted value functions arising from over-reliance on offline datasets. Reviewers commend the paper's originality in proposing a coherent algorithm that encapsulates solutions to these challenges. They highlight its clear presentation, supported by equations and well-structured experiments, emphasising its incremental change to the field of offline RL. However, concerns were raised regarding comparisons with off-policy algorithms utilising preload buffers, urging for deeper analyses of the challenges beyond proof-of-concept experiments, and questioning the computational cost of maintaining multiple policies. Additionally, reviewers suggested improvements in clarity, method description, and fair evaluation, specifically regarding the impact of evaluation rollouts on sample counts. Two reviewers also asked for comparison to CalQL

Post rebuttal,there is no clear consensus on the paper, with 2 reviewers learning towards rejection and the other two leaning towards acceptance. Post rebuttal, both reviewer uJfn and XFhn agree that although results look promising, they are not significant enough compared to prior methods (of note was CalQL). Moreover, the most positive reviewer DsmU, who also requested CalQL results, did not respond during the rebuttal period. Given that reviewer uJfn and XFhn were not impressed with these new results, and DsmU did not comment, then there is insufficient support for this paper to be accepted at this time.

**Justification For Why Not Higher Score:**

There is no clear consensus on the paper and marginal gain compared to prior methods.

**Justification For Why Not Lower Score:**

N/A

---

### Decision · Program_Chairs · 2024-01-16

Reject